# Pragmatic Heterogeneous Collaborative Perception via Generative Communication Mechanism

**Junfei Zhou**[1,2]    **Penglin Dai**[1,2*]    **Quanmin Wei**[1,2]    **Bingyi Liu**[3]
**Xiao Wu**[1,2]    **Jianping Wang**[4]

[1]Southwest Jiaotong University    [2]Engineering Research Center of Sustainable Urban Intelligent Transportation, Ministry of Education, China
[3]Wuhan University of Technology    [4]City University of Hong Kong
{jeffreychou, wqm}@my.swjtu.edu.cn    penglindai@swjtu.edu.cn
byliu@whut.edu.cn    wuxiaohk@gmail.com    jianwang@cityu.edu.hk

## Abstract

Multi-agent collaboration enhances the perception capabilities of individual agents through information sharing. However, in real-world applications, differences in sensors and models across heterogeneous agents inevitably lead to domain gaps during collaboration. Existing approaches based on adaptation and reconstruction fail to support *pragmatic heterogeneous collaboration* due to two key limitations: (1) Intrusive retraining of the encoder or core modules disrupts the established semantic consistency among agents; and (2) accommodating new agents incurs high computational costs, limiting scalability. To address these challenges, we present a novel **Gen**erative **Comm**unication mechanism (GenComm) that facilitates seamless perception across heterogeneous multi-agent systems through feature generation, without altering the original network, and employs lightweight numerical alignment of spatial information to efficiently integrate new agents at minimal cost. Specifically, a tailored Deformable Message Extractor is designed to extract spatial message for each collaborator, which is then transmitted in place of intermediate features. The Spatial-Aware Feature Generator, utilizing a conditional diffusion model, generates features aligned with the ego agent's semantic space while preserving the spatial information of the collaborators. These generated features are further refined by a Channel Enhancer before fusion. Experiments conducted on the OPV2V-H, DAIR-V2X and V2X-Real datasets demonstrate that GenComm outperforms existing state-of-the-art methods, achieving an 81% reduction in both computational cost and parameter count when incorporating new agents. Our code is available at https://github.com/jeffreychou777/GenComm.

## 1   Introduction

In autonomous driving field, multi-agent collaborative perception has emerged as a promising paradigm for enhancing environmental understanding by enabling information sharing among agents, thereby effectively extending perception range and mitigating challenges such as occlusions and long-range sensing limitations [1, 2, 3]. Recently, numerous studies have been conducted to advance this field [4, 5, 6, 7, 8]. Most of these works are based on the assumption of homogeneous collaboration, which limits their applicability in real-world scenarios, where collaboration typically involves heterogeneous agents with diverse sensor modalities and model architectures.

Existing approaches for heterogeneous multi-agent perception are predominantly non-generative and can be broadly categorized into two types: adaptation-based and reconstruction-based methods, illustrated in Figure 1 (a) and (b). Adaptation-based methods include using a single adapter for one-stage transformation such as MPDA[9], employing two-stage adaptation with a predefined

---

*Corresponding author.

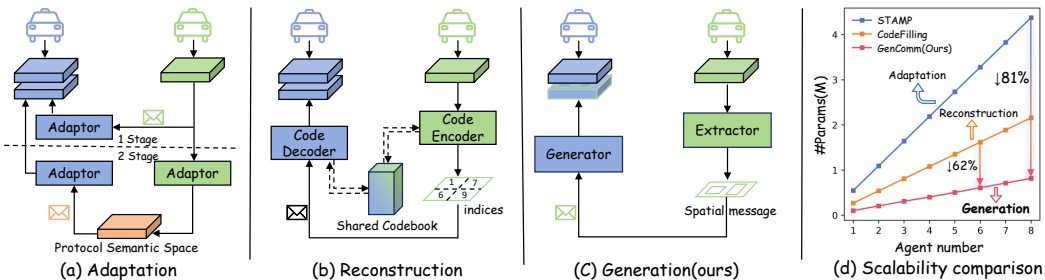

Figure 1: Comparison of heterogeneous collaboration strategies. (a) Adaptation-based strategy: transform features via one or two-stage adaptation. (b) Reconstruction-based strategy: reconstruct features on ego agent using indices of a shared codebook. (c) Ours (Generation-based): generate features locally using collaborators' spatial messages, requiring only lightweight extractor tuning to support new agents without modifying the core module. (d) Our method shows superior scalability over (a) and (b).

protocol semantic space such as PnPDA[10] and STAMP[11], or adopting the BackAlign strategy such as HEAL[12], which can be viewed as a special form of adaptation that enforces the collaborator's semantic space to align with the ego agent's semantic space. Reconstruction-based methods CodeFilling[13], on the other hand, reconstruct features locally using indexing of a shared codebook composed of encoded heterogeneous intermediate features. Despite their different mechanisms, both types of methods suffer from common limitations that fail to support *pragmatic heterogeneous collaborative perception*. For instance, PnPDA[10] and STAMP[11] rely on a predefined protocol semantic space, but the diversity of agents and vendors **makes reaching consensus on a protocol space unrealistic**. Similarly, methods like MPDA[9] and HEAL[12] require retraining the fusion network or encoders, which **disrupts the established semantic consistency among agents.** When accommodating new agents, these methods and CodeFilling **either require relatively high computational cost or introduce more parameters, thereby suffering from scalability constraints.** These shared limitations present fundamental barriers to the real-world application of collaborative perception systems in heterogeneous environments, highlighting *the core challenge of pragmatic heterogeneous collaborative perception: How can we accommodate the emerging new agents into the collaboration with minimal cost, while keeping the established semantic consistency among agents?*

To address these challenges, we propose GenComm, a **Gen**erative **Comm**unication mechanism for heterogeneous collaborative perception that facilitates seamless perception across heterogeneous multi-agent systems through feature generation, without altering the original network, and employs lightweight numerical alignment of spatial message to efficiently integrate new agents at minimal cost, shown in Figure 1 (c). The key idea behind GenComm is that each ego agent locally generates features for its collaborators using received spatial messages, ensuring that the generated features are aligned with the ego agent's semantic space while preserving the spatial information of its collaborators. To train the GenComm framework, initially conducted in a homogeneous setting, the model learns three key components: the Deformable Message Extractor, responsible for capturing spatial messages; the Spatial-Aware Feature Generator, aimed at generating features based on the received spatial messages from collaborators; and the Channel Enhancer, designed to refine the generated features along the channel dimension before fusion. Although the spatial messages shared among agents may exhibit a smaller domain gap compared to intermediate features, significant numerical discrepancies still arise due to inconsistent spatial confidence estimations across heterogeneous agents. Therefore, in the heterogeneous setting, each agent fine-tunes a lightweight message extractor specifically for its receiver to address the impact of numerical discrepancies across agents. Our method **meets all the requirements of *pragmatic heterogeneous collaborative perception* simultaneously** and reduces transmission volume, thereby improving communication efficiency.

Accordingly, our contributions can be summarized as follows:

- We propose the first generation-based communication mechanism called GenComm for heterogeneous collaborative perception, enabling seamless perception among heterogeneous agents through feature generation without altering the original network, while leveraging lightweight numerical alignment of spatial message to accommodate new agents at minimal cost. Additionally, it brings the benefit of improved communication efficiency.

- We design a Deformable Message Extractor to extract key spatial information, which is used as conditions for a spatial-aware feature generator that generates features for collaborators aligned with ego's semantic space and preserving the spatial information of collaborators, and a Channel Enhancer is designed to refine the generated feature at channel dimension before fusion.

- Extensive experiments on the OPV2V-H[12], DAIR-V2X[14] and V2X-Real[15] datasets demonstrate that GenComm outperforms state-of-the-art baselines in both simulated and real-world heterogeneous settings. Moreover, the cost of accommodating a new agent is reduced by over 81% and 62% compared to the leading adaptation- and reconstruction-based methods, respectively, highlighting its excellent scalability, shown in Figure 1 (d).

## 2 Related Works

**Collaborative perception.** Collaborative perception breaks through the limitations of single-vehicle sensing by extending the ego agent's perception capability to occluded and long-range regions. Among these, the most widely studied intermediate fusion approach[16, 17, 18, 19, 20, 21] enables individual agents to obtain more comprehensive features for downstream tasks by sharing and fusing intermediate features. Where2comm[4] and CodeFilling[13] have made efforts to balance communication overhead and performance, while methods such as CoAlign[5] and CoBevFlow[6] have been proposed to address feature misalignment caused by positional inaccuracies and temporal asynchrony. V2X-Radar[22] and V2X-R[23] incorporated radar data, exploiting the modality's robustness to adverse weather conditions to enhance the resilience of collaborative systems under challenging environments. While the above methods focus on enhancing collaborative perception under homogeneous settings, increasing attention has been given to heterogeneous collaboration, where differences in sensing modalities and model architectures introduce new challenges.

**Heterogeneous collaboration.** Many existing works have made significant efforts to address the challenges of heterogeneous collaboration. Both BM2CP [24] and HM-ViT [25] leverage modality-specific characteristics to enhance heterogeneous feature fusion. MPDA [9] adopts an adversarial domain adaptation framework to transform features in 1 stage. PnPDA [10] and STAMP [11] maintains a shared semantic space, enabling semantic feature transformation in 2 stage. HEAL[12] proposes a BackAlign strategy to align collaborator's semantic space to the ego agent's semantic space. CodeFilling [13] constructs a shared codebook by aggregating multi-domain features from all collaborators. Both feature transmission and reconstruction are performed through indexing into this shared codebook, which cleverly mitigates the domain gap among heterogeneous features by representing them in a unified latent space. Although progress has been made, existing methods still fall short of simultaneously meet the requirements of *pragmatic heterogeneous collaboration* due to either intrusive design and limited scalability.

**Feature generation.** Diffusion models [26, 27] are applied for Bird's-Eye-View (BEV) feature generation, leveraging their powerful denoising capabilities to enhance spatial representations. DiffBEV[28] further exploits the potential of diffusion models to generate more comprehensive BEV representations, where condition-guided diffusion sampling produces richer BEV features for downstream tasks. V2X-R [23] adopts robust 4D radar features as conditional inputs to denoise corrupted LiDAR features, resulting in cleaner representations. CoDiff [29] projects BEV features into a latent space and uses the projected representations as conditional guidance for diffusion-based sampling.

## 3 Pragmatic Heterogeneous Collaborative Perception

In a pragmatic multi-agent collaborative perception system, each agent collaborates with heterogeneous agents, while new agents continuously joining the collaboration. Assume there are currently $N$ agents, each equipped with a frozen pretrained perception network denoted as $\Psi_{\theta'}^i$. The method for heterogeneous collaboration is represented by a learnable module $\Phi_\theta^i$. When a new agent joins the system, increasing the number of agents to $N + 1$, it is essential to ensure that the integration incurs minimal parameter and computational overhead. This is crucial due to the limited computational capacity of each agent and the need to maintain scalability as more agents are integrated. The objective is to optimize the parameters $\theta$ of the heterogeneous collaboration module $\Phi_\theta^i$, such that the overall perception error and computational cost $\sum_i^{N+1} \text{Comp}(\Phi_\theta^i)$ of all $N + 1$ agents are minimized. Subject to the constraint that each agent's perception network $\Psi_{\theta'}^i$ remains fixed to preserve the

established semantic consistency among agents, the optimization problem is formulated as follows:

$$\min_{\theta} \quad \left( \sum_{i=1}^{N+1} d\left( \mathcal{O}_i, \Psi_{\theta'}^i \left( \mathcal{X}_i, \Phi_\theta^i \left( \{ \mathcal{M}_{j \to i} \}_{j \in \mathcal{G}_i} \right) \right) \right), \sum_{i=1}^{N+1} \text{Comp}(\Phi_\theta^i) \right)$$

$$\text{s.t.} \quad \theta_i' \in \Theta_{\text{frozen}}, \quad \forall i \in \{1, \ldots, N+1\} \tag{1}$$

Here, $\mathcal{O}_i$ denotes the ground-truth output for agent $i$, $\mathcal{M}_{j \to i}$ denotes the message extracted from agent $j$ to agent $i$, and the evaluation function $d(\cdot)$ measures the discrepancy between the ground-truth and the output predicted through collaboration, and $\mathcal{G}_i$ is the set of collaborators to agent $i$.

## 4    Methodology

In this section, we provide a detailed description of how the proposed GenComm is integrated into a multi-agent perception system. We further elaborate on the underlying principles and the operational workflow of this mechanism. Designed to enhance system scalability, preserve the the established semantic consistency among agents, and enable efficient message sharing, our approach provides a more practical solution for building pragmatic heterogeneous multi-agent collaborative perception systems.

### 4.1    Framework overview

In a practical multi-agent collaborative perception scenario, we consider $N$ heterogeneous agents, among which one is designated as the ego agent. For each agent $i$, we define $\mathcal{G}_i$ as the set of collaborators to ego agent $i$. Each agent $i \in \{1, 2, \ldots, N\}$ receives an input observation $\mathcal{X}_i$, which may consist of data from different sensing modalities such as images or LiDAR point clouds. To extract representations from these observations, each agent is equipped with an independent encoder $f_{\text{enc}}^i$, which transforms the inputs into BEV features $\mathcal{F}_i$.

Unlike previous methods that transmit full feature maps, our approach only requires the transmission of spatially-aware compressed representations. Specifically, after extracting features $\mathcal{F}_i$ using a dedicated encoder, each agent utilizes a Deformable Message Extractor $f_{\text{mes\_extract}}^{i \to j}$ to capture a spatial message, which is then combined with the agent's meta information for transmission. Upon receiving messages from collaborators, the ego agent uses them as conditional inputs to generate features that align to its own semantic space while preserving spatial information from the collaborators. A Channel Enhancer refines the generated features along the channel dimension before fusion. A fusion network is employed to integrate the ego features with the generated collaborator features, aiming to obtain more comprehensive global representations. The fused feature is finally passed to the decoder to obtain perception results. The following provides the mathematical representation:

$$\mathcal{F}_i = f_{\text{enc}}^i(\mathcal{X}_i) \tag{2a}$$

$$\mathcal{M}_{j \to i} = f_{\text{mes\_extract}}^{j \to i}(\mathcal{F}_i), \quad \forall j \in \mathcal{G}_i \tag{2b}$$

$$\{\hat{\mathcal{F}}_j\}_{j \in \mathcal{G}_i} = f_{\text{generate}}^i \left( \mathcal{F}_i, \{\mathcal{M}_{j \to i}\}_{j \in \mathcal{G}_i} \right) \tag{2c}$$

$$\mathcal{Z}_i = f_{\text{fusion}}^i \left( \mathcal{F}_i, f_{\text{ch\_enhance}}^i \left( \{\hat{\mathcal{F}}_j\}_{j \in \mathcal{G}_i} \right) \right) \tag{2d}$$

$$\hat{\mathcal{O}}_i = f_{\text{dec}}^i(\mathcal{Z}_i) \tag{2e}$$

where $\mathcal{F}_i \in \mathbb{R}^{C_i \times H_i \times W_i}$ denotes the BEV feature of the ego agent $i$, with $C_i$, $H_i$, and $W_i$ representing the channel, height, and width dimensions, respectively. $\mathcal{M}_{j \to i}$ denotes the message extracted from agent $j$ to agent $i$. The set $\{\hat{\mathcal{F}}_j\}_{j \in \mathcal{G}_i}$ represents the features generated by agent $i$ for its collaborators. $\mathcal{Z}_i$ is the fused feature of agent $i$, and $\hat{\mathcal{O}}_i$ denotes the final output obtained after decoding.

### 4.2    Model components

An overview of the system is presented in Figure 2, followed by a detailed description of the three key components of GenComm. Additional detailed information about the model is provided in the Appendix A.3.

**Deformable message extractor.** Extracting generalizable spatial information from features of different modalities processed by distinct encoders is a critical component of our approach. The extracted spatial information serves not only as the transmitted message for inter-agent communication but

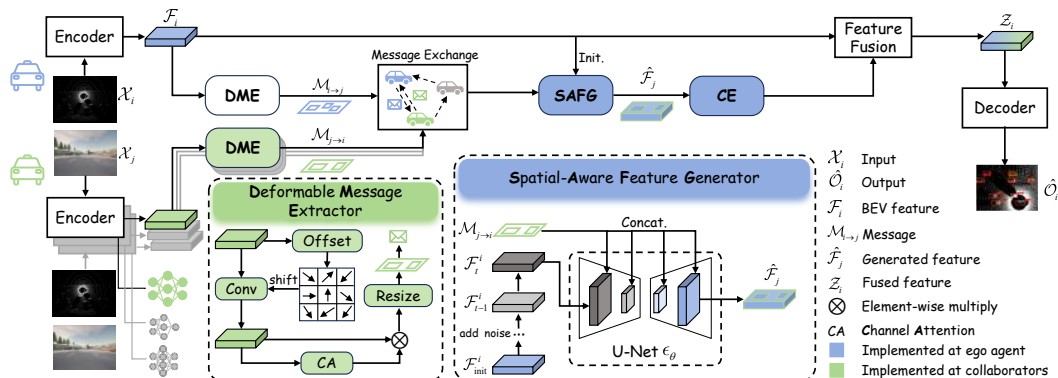

Figure 2: Framework overview of GenComm. The Deformable Message Extractor extracts spatial messages, which serve as conditions for the Spatial-Aware Feature Generator. A Channel Enhancer further refines the generated features before fusion.

also as spatial condition for the ego agent's feature generation. The quality of this spatial information directly determines the spatial fidelity of the generated features.

Intuitively, spatial information relevant to 3D object detection tasks can be represented by the confidence at individual pixels of BEV features. However, confidence at a single pixel is insufficient to capture foreground-background separation and contour information within BEV features. Therefore, we design a Deformable Message Extractor incorporating deformable convolution[30], which not only focuses on the pixel itself but also dynamically references surrounding pixels, thereby enhancing the model's capability to distinguish between foreground and background.

Formally, for an input feature map $\mathcal{F}_i \in \mathbb{R}^{C_i \times H_i \times W_i}$, an offset prediction network predicts a set of sampling offsets $\Delta \mathbf{p}_{i,k} \in \mathbb{R}^{2K \times H_i \times W_i}$. Given these offsets, spatial information is extracted using a deformable convolution defined as:

$$\mathcal{M}_{i \to j} = \text{Resizer} \left( \sum_{\mathbf{p}_k \in \mathcal{R}} w(\mathbf{p}_k) \cdot \mathcal{F}_i \left( \mathbf{p}_0 + \mathbf{p}_k + \Delta \mathbf{p}_{i,k} \right) \right) \in \mathbb{R}^{C' \times H_j \times W_j} \tag{3}$$

Where $\mathcal{R}$ denotes the regular convolution sampling grid, $k$ is the kernel size, and $w(\cdot)$ represents the learnable attention weights. To guarantee compatibility across agents, a learnable resizer is introduced to adapt the spatial information to the resolution of the receiver. In this context, $C'$ denotes the compressed channel dimension, while $\mathcal{M}_{i \to j}$ corresponds to the extracted spatial message that serves as the basis for downstream communication and generation.

**Spatial-aware feature generator.** The ego agent $i$ upon receiving spatial messages $\{\mathcal{M}_{j \to i}\}_{j \in \mathcal{G}_i}$ from collaborators, then leverages a conditional diffusion model[31] to generate feature representations that align with its own semantic space while preserving collaborator's spatial information.

Gaussian noise is progressively added to the initial feature $\mathcal{F}_{\text{init}}$ (where initialization from $\mathcal{F}_i$) over $\mathcal{T}$ time steps. Let $\mathcal{F}_t$ denote the feature at the denoising step $t$, then the diffusion process $q$ can then be expressed as:

$$q(\mathcal{F}_t \mid \mathcal{F}_{\text{init}}) = \mathcal{N} \left( \mathcal{F}_t \mid \sqrt{\bar{\alpha}_t} \mathcal{F}_{\text{init}}, (1 - \bar{\alpha}_t) \mathbf{I} \right) \tag{4}$$

Specifically, given the initial feature $\mathcal{F}_{\text{init}}$, the noisy feature $\mathcal{F}_t$ at timestep t can be directly obtained via the closed-form of the forward diffusion process:

$$\mathcal{F}_t = \sqrt{\bar{\alpha}_t} \, \mathcal{F}_{\text{init}} + \sqrt{1 - \bar{\alpha}_t} \, \mathbf{z}, \quad \mathbf{z} \sim \mathcal{N}(0, \mathbf{I}) \tag{5}$$

This perturbed feature $\mathcal{F}_t$ serves as the input to the denoising process, where a conditional U-Net $\epsilon_\theta$ generates the feature representation at the previous timestep. At each generation step, the received message $\{\mathcal{M}_{j \to i}\}_{j \in \mathcal{G}_i}$ is incorporated as a conditioning input, guiding the U-Net $\epsilon_\theta$ to generate features:

$$\{\mathcal{F}_{t-1}^j\}_{j \in \mathcal{G}_i} = \epsilon_\theta \left( [\mathcal{F}_t \,\|\, \{\mathcal{M}_{j \to i}\}_{j \in \mathcal{G}_i}], t \right), \quad t \in \mathcal{T}, \mathcal{T} - 1, \dots, 1 \tag{6}$$

After the final generation step $\mathcal{T}$, the model generates features $\{\hat{\mathcal{F}}_j\}_{j \in \mathcal{G}_i}$ that are aligned with the semantic space of ego agent $i$, while preserving the spatial information received from collaborators.

The generation process is directly supervised with the objective $\mathcal{L}_{\text{GEN}}$ defined as the mean squared error between the generated features and their corresponding ground truth:

$$\mathcal{L}_{gen} = \sum_{j \in \mathcal{G}_i} \left\| \hat{\mathcal{F}}_j - \mathcal{F}_j \right\|_2^2 \tag{7}$$

This design allows the ego agent to generates features aligned with the ego agent's semantic space while preserving the spatial information of the collaborators.

**Channel enhancer.** During the feature generation process, spatial information typically dominates, while semantic representations along the channel dimension are often overlooked. To ensure consistency between the ego features and the generated features in the channel dimension, we propose the Channel Enhancer. Specifically, we introduce the PConv operation[32] to enhance the representational capacity of informative elements within the features. A gating mechanism[33] is incorporated to suppress redundant information along channel dimension, while channel attention is applied to emphasize critical features.

Firstly, we adopt PConv to reinforce the feature, then splits the reinforced feature into two parts along the channel dimension: a modifiable part $\mathcal{F}_{\text{conv}}$ and a static part $\mathcal{F}_{\text{res}}$. The modifiable part is refined using a depthwise separable convolution and multiply by $\mathcal{F}_{\text{res}}$. Following channel refinement, we apply a channel-wise attention mechanism to suppress redundant channels and highlight informative ones. We compute a soft attention score based on global context, the final refined feature is obtained. This module improves the channel-wise expressiveness of the generated features, ensuring better alignment with ego-agent representations:

$$\mathcal{F}' = \sigma \left( \text{LN} \left( \text{Conv}_{\text{pw}} \left( \text{Conv}_{\text{dw}}(\mathcal{F}_{\text{conv}}) \right) \odot \mathcal{F}_{\text{res}} \right) \right) \tag{8a}$$

$$\tilde{\mathcal{F}}_i = \mathcal{F}' \odot \text{Softmax}(\text{FC}_2(\sigma(\text{LN}(\text{FC}_1(\text{GAP}(\mathcal{F}')))))) \tag{8b}$$

Here, $\odot$ denotes element-wise multiplication; GAP stands for global average pooling; FC denotes a fully connected layer; LN represents a linear transformation; and $\sigma$ is the activation function.

### 4.3 Training strategy

The training process of GenComm consists of two stages. In the first stage, the model is trained in a homogeneous multi-agent perception setting, where each agent constructs its own semantic space and learns three key components: a Deformable Message Extractor, a Spatial-Aware Feature Generator and a Channel Enhancer. After this first stage, the Deformable Message Extractor effectively captures spatial information, helping to avoid the domain gap issues commonly associated with intermediate features. However, it still suffers from numerical discrepancies, such as inconsistent confidence scores for the same pixels in the BEV map across heterogeneous agents. To mitigate this, in the second stage, each agent is initialized with a lightweight extractor tailored to the specific agent with which it will collaborate. Only the extractor is fine-tuned to align the spatial message, ensuring that the data received by the receiver is more consistent in numerical distribution with its own extracted spatial information. This alignment enhances the quality of the subsequently generated features.

### 4.4 Loss function

In Stage 1, we train our model in an end-to-end manner using a combination of losses: a focal loss[34] for the classification loss $\mathcal{L}_{\text{cls}}$, a smooth L1 loss[35] for the regression loss $\mathcal{L}_{\text{reg}}$, and a mean squared error (MSE) loss for the generation loss $\mathcal{L}_{\text{gen}}$. The classification and regression losses are standard 3D object detection losses, used to predict the confidence of each anchor and to regress the offset between the anchor and the ground truth object. Each loss is weighted by a corresponding parameter $\alpha$ to balance its contribution, and the total loss is defined as $\mathcal{L}_{\text{stage1}} = \alpha_1 \mathcal{L}_{\text{cls}} + \alpha_2 \mathcal{L}_{\text{reg}} + \alpha_3 \mathcal{L}_{\text{gen}}$. In Stage 2, only the classification and regression losses are applied, resulting in $\mathcal{L}_{\text{stage2}} = \alpha_1 \mathcal{L}_{\text{cls}} + \alpha_2 \mathcal{L}_{\text{reg}}$.

## 5 Experimental Result

In this section, we evaluate the effectiveness of our approach using both simulated and real-world datasets, showing that our method delivers superior performance while integrating new agents at an ultra-low cost. Furthermore, we validate the contribution of each framework component through ablation studies.

Table 1: Performance on OPV2V-H and DAIR-V2X under various heterogeneous settings. AP is used to evaluate detection performance, while communication volume measures communication efficiency.

| Fusion Network | Method | OPV2V-H | | | | DAIR-V2X | | Comm. Volume $(log_2)\downarrow$ |
|---|---|---|---|---|---|---|---|---|
| | | $\mathbf{L}_P^{64}$-$\mathbf{L}_S^{32}$ | | $\mathbf{L}_P^{64}$-$\mathbf{C}_E$ | | $\mathbf{L}_P^{64}$-$\mathbf{L}_S^{40}$ | | |
| | | AP50 ↑ | AP70 ↑ | AP50 ↑ | AP70 ↑ | AP30 ↑ | AP50 ↑ | |
| AttFuse[16] | MPDA[9] | 0.7668 | 0.5698 | 0.7369 | 0.5739 | 0.4246 | 0.3641 | 22.0 |
| | BackAlign[12] | 0.7873 | 0.5841 | 0.6852 | 0.5240 | 0.4562 | 0.3727 | 22.0 |
| | CodeFilling[13] | 0.7218 | 0.5364 | 0.6661 | 0.5097 | 0.3848 | 0.3189 | **15.0** |
| | STAMP[11] | 0.7594 | 0.5689 | 0.7258 | 0.5605 | 0.4468 | **0.3913** | 22.0 |
| | GenComm | **0.8043** | **0.6332** | **0.7525** | **0.6005** | 0.4593 | 0.3786 | 16.0 |
| V2X-ViT[17] | MPDA[9] | 0.8495 | 0.6595 | 0.6873 | 0.5023 | 0.4717 | 0.3786 | 22.0 |
| | BackAlign[12] | 0.8553 | 0.6933 | 0.6907 | 0.5232 | 0.4902 | 0.3924 | 22.0 |
| | CodeFilling[13] | 0.8597 | 0.6886 | 0.5600 | 0.4156 | 0.4446 | 0.3557 | **15.0** |
| | STAMP[11] | 0.8442 | 0.6276 | 0.7508 | 0.5442 | 0.5421 | **0.4935** | 22.0 |
| | GenComm | **0.8673** | **0.6991** | **0.7630** | **0.5759** | **0.5651** | 0.4665 | 16.0 |

## 5.1 Experiments setting

**Datasets.** We conduct experiments on three datasets: OPV2V-H[12], DAIR-V2X[14] and V2X-Real[15]. OPV2V-H is an extension of the large-scale OPV2V[16] dataset, which was collected using the OpenCDA[36] framework in the CARLA simulator[37]. OPV2V-H expands it by incorporating additional sensor configurations, making it well-suited for heterogeneous collaborative perception research. DAIR-V2X is the first real-world dataset, the RSU in it is equipped with a high-resolution 300-channel LiDAR and a camera, while the vehicle has a 40-channel LiDAR and a camera, enabling exploration of cross-agent collaboration under heterogeneous sensor setups. V2X-Real is also a real-world dataset but on a larger scale, consisting of 2 vehicles and 2 RSUs (4 agents in total), which can be used to evaluate scalability in real-world scenarios.

**Heterogeneous agents.** Following prior work[11][12], four agents are considered on OPV2V-H: two equipped with LiDAR and two with cameras. For the LiDAR-based agents, we adopt PointPillars[38] and SECOND[39] as the respective backbones, denoted as $\mathbf{L}_P^{64}$ and $\mathbf{L}_S^{32}$, where the superscript indicates the channel size of the LiDAR sensor. For the camera-based agents, EfficientNet and ResNet[40] are used as backbones, denoted as $\mathbf{C}_E$ and $\mathbf{C}_R$, respectively. For the V2X-Real dataset, we implement four agents with backbones of varying capacities (ranging from shallow to deep) and employ the PointPillar encoder for all agents, aiming to investigate collaboration across different levels of feature extraction capability. The detailed network architectures and configurations are provided in the Appendix A.3. The substantial differences among these agents ensure the diversity and richness of our experimental setup.

**Baselines.** We incorporate several adaptation-based methods, including the one-stage method MPDA[9] and the two-stage method STAMP[11], as well as BackAlign from HEAL[12],which enforces the alignment of collaborators' semantic spaces to the ego agent's semantic space. MPDA requires retraining both the fusion network and the task head, whereas BackAlign retrains the encoder to achieve feature alignment. STAMP introduces an additional protocol semantic space to facilitate 2 stage adaptation. In addition, we include the reconstruction-based method CodeFilling[13], which reconstructs collaborator features using a shared codebook and the received indices.

**Implementation details.** For all baseline methods, we first train a base model for each agent in a homogeneous scenario, and then extend it to heterogeneous collaboration using their respective adaptation strategies. For our proposed GenComm, we train the entire system end-to-end in the homogeneous setting, and enable heterogeneous collaboration by aligning a lightweight Message Extractor for each newly introduced collaborator with respect to the ego agent. All methods, including baselines and GenComm, are trained under the same settings for fair comparison on NVIDIA RTX 3090. Implementation procedures and specific configurations are detailed in the Appendix A.3.

## 5.2 Quantitative results

**Heterogeneous Collaboration.** We evaluate our proposed method on three datasets encompassing both static and dynamic heterogeneous collaboration scenarios, covering two sensing modalities, four distinct encoder architectures, and two classical fusion method. In static collaboration scenarios,

Table 2: Performance comparison of different methods on two datasets as more agents are incorporated into collaboration. #P and #F denote the trained parameters and FLOPs, respectively.

**OPV2V-H**

| Method | $\mathbf{L}_P^{128}+\mathbf{C}_E$ | | $\mathbf{L}_P^{128}+\mathbf{C}_E+\mathbf{L}_S^{32}$ | | $\mathbf{L}_P^{128}+\mathbf{C}_E+\mathbf{L}_S^{32}+\mathbf{C}_R$ | | #P(M)↓ | #F(G)↓ |
|---|---|---|---|---|---|---|---|---|
| | AP50 ↑ | AP70 ↑ | AP50 ↑ | AP70 ↑ | AP50 ↑ | AP70 ↑ | | |
| MPDA[9] | 0.7574 | 0.5497 | 0.6513 | 0.4786 | 0.6815 | 0.5123 | 5.75 | 51.93 |
| BackAlign[12] | 0.6975 | 0.5288 | 0.7238 | 0.5398 | 0.7252 | 0.5408 | 31.18 | 211.38 |
| CodeFilling[13] | 0.6891 | 0.5234 | 0.637 | 0.4658 | 0.5981 | 0.4316 | 0.81 | 12.91 |
| STAMP[11] | **0.7609** | 0.5878 | 0.7819 | 0.5995 | 0.7829 | 0.6002 | 1.64 | 3.084 |
| GenComm | 0.7538 | **0.5951** | **0.7873** | **0.6174** | **0.7866** | **0.6184** | **0.31** | **0.615** |

**V2X-Real**

| Method | $\mathbf{L}_H^{128}+\mathbf{L}_L^{128}$ | | $\mathbf{L}_H^{128}+\mathbf{L}_L^{128}+\mathbf{L}_M^{128}$ | | $\mathbf{L}_H^{128}+\mathbf{L}_L^{128}+\mathbf{L}_M^{128}+\mathbf{L}_T^{128}$ | | #P(M)↓ | #F(G)↓ |
|---|---|---|---|---|---|---|---|---|
| | AP30 ↑ | AP50 ↑ | AP30 ↑ | AP50 ↑ | AP30 ↑ | AP50 ↑ | | |
| MPDA[9] | 0.6344 | 0.5725 | 0.6323 | 0.5751 | 0.6211 | 0.5672 | 5.75 | 51.93 |
| BackAlign[12] | 0.6313 | 0.5822 | 0.6386 | 0.5878 | 0.6352 | 0.5896 | 31.18 | 211.38 |
| CodeFilling[13] | 0.6273 | 0.5826 | 0.6284 | 0.5799 | 0.6081 | 0.5571 | 0.81 | 12.91 |
| STAMP[11] | 0.6314 | 0.5881 | 0.6335 | 0.5893 | 0.6289 | 0.5882 | 1.64 | 3.084 |
| GenComm | **0.6848** | **0.6175** | **0.6961** | **0.6299** | **0.7144** | **0.6362** | **0.31** | **0.615** |

Figure 3: Comparision of performance with baselines on the pose error and time delay setting.

as shown in Table 1, our method outperforms the baseline methods in the majority of cases while reducing communication overhead by up to 64×. The performance improvement primarily stems from the tailored Deformable Message Extractor, Spatial-Aware Feature Generator and Channel Enhancer, which jointly enable precise spatial information extraction, high-quality feature generation, and rich semantic preservation—effectively narrowing the semantic gap across heterogeneous agents. In dynamic collaboration scenarios, illustrated in Table 2, we evaluate all methods on the OPV2V-H and V2X-Real datasets. In this setting, more agents progressively join the collaboration. Notably, GenComm, BackAlign, and STAMP maintain consistent performance gains as the number of collaborators increases, whereas MPDA and CodeFilling do not. Moreover, GenComm achieves the highest performance with the minimal computational and parameter cost. Importantly, only STAMP and our approach are non-intrusive during heterogeneous collaboration.

**Scalability analysis.** We analyze the scalability of our method in Table 2 and Figure 1 (d). Thanks to the lightweight numerical alignment mechanism, our approach can continuously accommodate newly added collaborators with minimal computational overhead. In Table 2, we designate $\mathbf{L}_p^{64}$ as the ego agent and incrementally introduce three heterogeneous agents for collaboration. Our method not only outperforms the baselines in terms of accuracy, but also reduces parameter and computation costs by 80% compared to the latest state-of-the-art methods STAMP. This demonstrates that our method maintains extremely low incremental cost as more agents join the collaboration, highlighting its strong scalability.

**Robustness analysis.** In real-world applications, pose errors and time delays are as inevitable as heterogeneous collaboration. To better assess the practicality of our method, we further evaluate its robustness under pose errors and time delays. Specifically, we introduce Gaussian noise and asynchrony input to simulate pose errors and time delays. Experimental results demonstrate that our method exhibits superior robustness to both pose errors and time delay compared to state-of-the-art approaches shown in Figure 3.

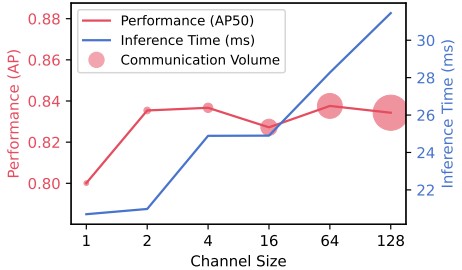

Table 3: Ablation Study on Message Extractor, Channel Enhance, and Extractor Align

| DME | CE | Align | AP50 ↑ | AP70 ↑ |
|-----|----|----|----|----|
| | | | 0.2850 | 0.1922 |
| | ✓ | | 0.7555 | 0.5961 |
| ✓ | | | 0.7805 | 0.4911 |
| ✓ | | ✓ | 0.7877 | 0.5480 |
| ✓ | ✓ | | 0.7514 | 0.5709 |
| ✓ | ✓ | ✓ | **0.8043** | **0.6332** |

Figure 4: Ablation study on the channel size of the spatial message.

**Ablation study.** We conduct ablation studies to analyze the effectiveness of the proposed components, as shown in Table 3 and Figure 4, which present the results of component-wise ablation and channels of message ablation, respectively. The results indicate that the Channel Enhancer plays a crucial role in refining the generated features, while the Deformable Message Extractor effectively captures accurate spatial information to guide the generation process. Additionally, the lightweight numeric alignment helps mitigate spatial message discrepancies across agents. As illustrated in Figure 4, we conduct an ablation study on the channel size of the spatial message to investigate the trade-off among model performance, communication volume, and inference time. The red line denotes the model performance, the blue line represents the inference time of the diffusion model, and the red circles indicate the corresponding communication cost. Overall, our message design achieves a balanced compromise across these three factors.

## 5.3 Visualization

The visualization results demonstrate that the semantic space of the generated features is consistent with that of the ego agent, while fully preserving the spatial information of the collaborators. In terms of both feature quality and detection performance, our method surpasses CodeFilling.

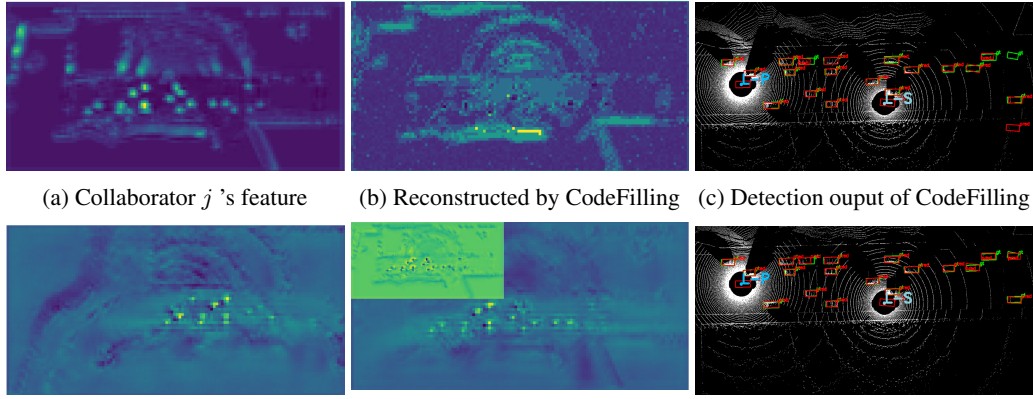

(a) Collaborator $j$'s feature    (b) Reconstructed by CodeFilling    (c) Detection ouput of CodeFilling

(d) Semantic space of ego agent $i$    (e) $\mathcal{M}_{j \to i}$ & Generated feature $\hat{\mathcal{F}}_j$    (f) Detection ouput of GenComm

Figure 5: Visualization result of our generation method, Compared to CodeFilling, generated feature $\hat{\mathcal{F}}_i$ is more consistent with the ego agent $i$'s semantic space and preserving the spatial information from collaborator $j$. The prediction and GT shown in red and green respectively.

## 6 Solution for Real-World Application

In real-world deployments, we assume three vendors: $A, B, C$, and five heterogeneous agent types: $A_1, A_2, B_1, B_2, B_3, C$, where each agent type refers to a specific combination of sensor and model architecture. Below we describe how GenComm can be practically applied in such scenarios:

**Stage 1: Homogeneous Pre-training.** Each vendor trains their agents using GenComm in a homogeneous collaboration setting. For example, vendor $A$ trains agent $A_1$ in collaboration with other instances of $A_1$ using their **private data** belonging to vendor $A$ **independently**. In this stage, the fu-

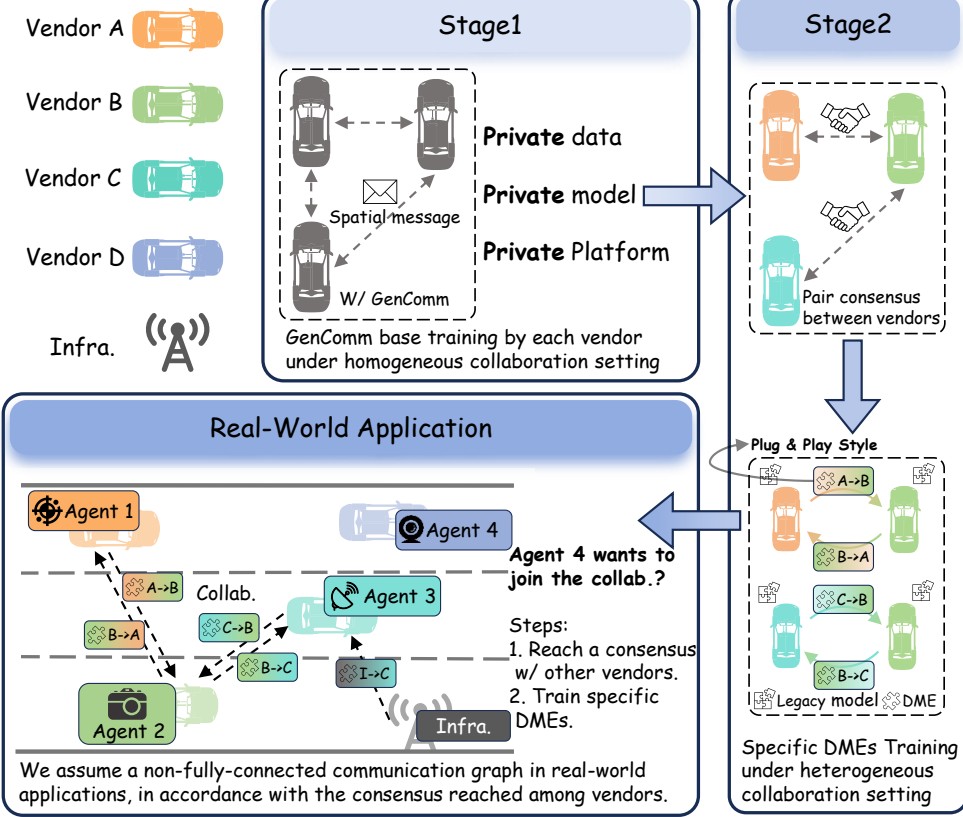

Figure 6: Application rationale of GenComm.

sion network, generation module, and Channel Enhancer are trained effectively under a homogeneous scenarios.

**Stage 2: Heterogeneous Collaboration.** If two vendors reach a collaboration consensus, they can enable heterogeneous collaboration by training specific Deformable Message Extractors (DMEs) between heterogeneous agents. For instance, if vendors A and B agree to collaborate, they train DMEs such as: $DME_{A_1 \rightarrow B_1}$, $DME_{A_1 \rightarrow B_2}$, ..., $DME_{B_3 \rightarrow A_2}$.

These lightweight modules enable heterogeneous collaboration by deploying corresponding DMEs to each agent in a **plug-and-play, non-intrusive** style. When new agents join, vendors only need to deploy the corresponding DMEs without further modifications, thereby preserving the established semantic consistency among agents.

## 7 Conclusion

We propose GenComm, a generative communication mechanism designed to support *pragmatic heterogeneous collaborative perception*. GenComm facilitates seamless perception across heterogeneous multi-agent systems through feature generation, without altering the original network, and employs lightweight numerical alignment of spatial information to efficiently integrate new agents at minimal cost. Comprehensive experimental results demonstrate that GenComm not only achieves strong performance, but also exhibits excellent scalability and communication efficiency. These advantages make GenComm a potential solution for large-scale deployment in real-world multi-agent systems, contributing to enhanced safety in autonomous driving and a reduced risk of accidents.

**Limitations.** Although we assume a more realistic non-fully-connected communication graph in our application rational, the approach still requires consensus among vendors, which may be hindered by factors such as commercial competition and the potential risks of malicious attacks.

**Acknowledgment.** This work was supported in part by the National Natural Science Foundation of China under Grant 62172342, Grant 62372387; Key R&D Program of Guangxi Zhuang Autonomous Region, China (Grant No. AB22080038, AB22080039); The Open Fund of the Engineering Research Center of Sustainable Urban Intelligent Transportation, Ministry of Education, China (Project No. KCX2024-KF07).

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

# A  Appendix

## A.1  Advantages in real-world applications

In Table A.1, we compare several heterogeneous collaboration methods with our proposed generative approach in terms of their characteristics and capabilities. Specifically, "Multi-Mod." and "Multi-Enc." indicate whether each method demonstrates the effectiveness of multi-modality and multi-encoder settings, respectively. "Non-Intru." refers to whether the pre-trained models or core modules remain non-intrusive. "Scal." denotes scalable, indicating whether the method can seamlessly accommodate newly introduced agents at minimal cost. "Comm. Eff." reflects whether the approach is communication-efficient. These merits are crucial for the real-world applications of multi-agent systems. Among all existing methods, only GenComm simultaneously satisfies all these merits, which demonstrates that our method is more practical than prior approaches and highlights the superiority of our proposed framework.

Table A.1: Comparison of methods on key properties.

| Method | Publication | Multi-Mod. | Multi-Enc. | Non-Intru. | Scal. | Comm. Eff. |
|---|---|---|---|---|---|---|
| MPDA[9] | ICRA 2023 | | ✓ | | | |
| HM-ViT[25] | ICCV 2023 | ✓ | | | | |
| HEAL[12] | ICLR 2024 | ✓ | ✓ | | | |
| CodeFilling[13] | CVPR 2024 | ✓ | | | | ✓ |
| PnPDA[10] | ECCV 2024 | | ✓ | ✓ | | |
| STAMP[11] | ICLR 2025 | ✓ | ✓ | ✓ | ✓ | |
| GenComm | - | ✓ | ✓ | ✓ | ✓ | ✓ |

## A.2  Algorithmic Description of GenComm

Here, we present the algorithmic pipeline of the proposed method in Algorithm 1, which summarizes the key steps and provides a clear overview of the overall mechanism.

---

**Algorithm 1:** GenComm: Overall Algorithmic Pipeline

---

**Data:** $N$: total number of collaborative agents; $\mathcal{G}_i$: set of collaborators for agent $i$
**Input:** $\mathcal{X}_i$: observation of agent $i$
**Output:** $\hat{\mathcal{O}}_i$: prediction of agent $i$
**for** $i = 1$ **to** $N$ **do**
 $\mathcal{F}_i = f_{\text{enc}}^i(\mathcal{X}_i) \in \mathbb{R}^{C_i \times H_i \times W_i}$       // ▷ Feature extraction
 $\mathcal{M}_{i \to j} = f_{\text{mes\_extrac}}^{i \to j}(\mathcal{F}_i) \in \mathbb{R}^{C_j \times H_j \times W_j}, \quad \forall j \in \mathcal{G}_i$   // ▷ Message extraction
**end**
# The lightweight messages $\mathcal{M}_{i \to j}$ are used in place of raw features $\mathcal{F}_i$ for communication.
**for** $i = 1$ **to** $N$ **do**
 $\mathcal{F}_{\text{init}} \leftarrow \mathcal{F}_i$            // ▷ Initialization
 $\mathbf{z} \sim \mathcal{N}(0, \mathbf{I})$
 $\mathcal{F}_t = \sqrt{\bar{\alpha}_t}\,\mathcal{F}_0 + \sqrt{1 - \bar{\alpha}_t}\,\mathbf{z}$      // ▷ Diffusion forward process
 **for** $t \leftarrow \mathcal{T}, \mathcal{T} - 1, \ldots, 0$ **do**
  $\{\mathcal{F}_{t-1}^j\}_{j \in \mathcal{G}_i} = \epsilon_\theta([\mathcal{F}_t \,\|\, \{\mathcal{M}_{j \to i}\}_{j \in \mathcal{G}_i}], t)$   // ▷ Feature generation
 **end**
 $\{\hat{\mathcal{F}}_j\}_{j \in \mathcal{G}_i} \leftarrow \{\mathcal{F}_{\text{end}}^j\}_{j \in \mathcal{G}_i}$
 $\{\tilde{\mathcal{F}}_j\}_{j \in \mathcal{G}_j} = f_{\text{ch\_enhance}}^i(\{\hat{\mathcal{F}}_j\}_{j \in \mathcal{G}_i})$    // ▷ Channel enhancement
 $\mathcal{Z}_i = f_{\text{fusion}}^i(\mathcal{F}_i, \{\tilde{\mathcal{F}}_j\}_{j \in \mathcal{G}_j})$     // ▷ Feature fusion
 $\hat{\mathcal{O}}_i = f_{\text{dec}}^i(\mathcal{Z}_i)$
**end**

---

## A.3  Implementation details

### A.3.1  Component details

Figure A.1 illustrates the detailed workflow of the Channel Enhancer module. Here, PConv denotes Partial Convolution[32], dwconv denotes Depth-wise Convolution, pwconv stands for Point-wise Convolution, and LN represents a Linear Layer. The symbol $\odot$ indicates element-wise multiplication.

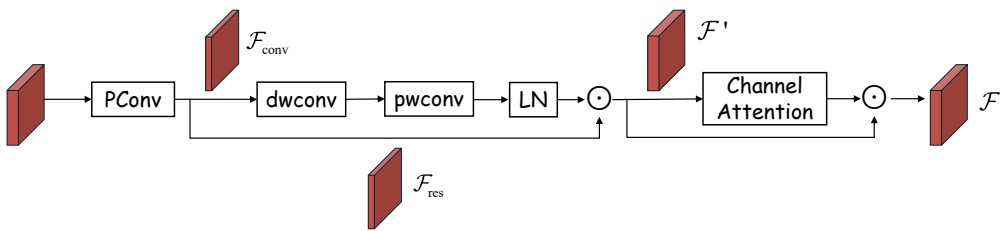

Figure A.1: Details of the channel enhancer

Table A.2: Configuration of the four heterogeneous agents

|         | Symbol | Sensor | Encoder | Backbone | #Params (M) |
|---------|--------|--------|---------|----------|-------------|
| OPV2V-H & DAIR-V2X | | | | | |
| Agent 1 | $\mathbf{L}_P^{64}$ | 64-Channel Lidar | PointPillar[38] | Deep | 6.58 |
| Agent 2 | $\mathbf{C}_E$ | RGB Camera | EfficientNet[41] | Deep | 21.25 |
| Agent 3 | $\mathbf{L}_S^{32}$ | 32-Channel Lidar | SECOND[39] | Shallow | 0.89 |
| Agent 4 | $\mathbf{C}_R$ | RGB Camera | Resnet101[40] | Deep | 8.15 |
| V2X-Real | | | | | |
| Agent 1 | $\mathbf{L}_H^{128}$ | 128-Channel Lidar | PointPillar[38] | Deep | 8.07 |
| Agent 2 | $\mathbf{L}_L^{128}$ | 128-Channel Lidar | PointPillar[38] | Medium | 2.23 |
| Agent 3 | $\mathbf{L}_M^{128}$ | 128-Channel Lidar | PointPillar[38] | Shallow | 1.06 |
| Agent 4 | $\mathbf{L}_T^{128}$ | 128-Channel Lidar | PointPillar[38] | Identity[*] | 0.76 |

[*] Identity indicates that the backbone is implemented as `nn.Identity()`.

Channel Attention refers to applying an attention mechanism along the channel dimension, followed by weighting the features with the corresponding attention scores to obtain the enhanced feature representation.

### A.3.2 Heterogeneous agents details

For the OPV2V-H and DAIR-V2X datasets, four heterogeneous agents are designed by combining three types of sensor data, four encoder architectures, and two backbone networks, supporting heterogeneous collaboration, denoted as $\mathbf{L}_P^{64}$, $\mathbf{C}_E$, $\mathbf{C}_R$ and $\mathbf{L}_S^{32}$, shown in Table A.2. Specifically, the sensing modalities include a 64-channel LiDAR, a 32-channel LiDAR, and an RGB camera. Point cloud data are processed using two different encoders: PointPillars[38] and SECOND[39]. Image features are extracted using EfficientNet[41] and ResNet-101[40]. For the V2X-Real dataset, four agents are designed with varying feature extraction capacities, ranging from deep to shallow backbones and even without a backbone. All agents adopt the PointPillar encoder and are denoted as $\mathbf{L}_H^{128}$, $\mathbf{L}_L^{128}$, $\mathbf{L}_M^{128}$, and $\mathbf{L}_T^{128}$.

In these agents, BEV Backbones of different scales are employed to extract spatial features. The shallow backbone consists of a single block with 128 output channels, designed to retain fine-grained spatial information at minimal computational cost. The medium backbone includes two blocks with 256 output channels, providing a balance between efficiency and representational capacity. The deep backbone adopts three blocks with 384 output channels, offering enhanced feature extraction capability.

### A.3.3 Experimental setting details

**Collaborative perception settings.** In our collaborative perception setting, the maximum communication range is set to 70 m. During training, the perception range for LiDAR-equipped agents is set to $[-102.4\,\mathrm{m},\ 102.4\,\mathrm{m}]$ along the $x$-axis and $[-51.2\,\mathrm{m},\ 51.2\,\mathrm{m}]$ along the $y$-axis, covering a total area of $204.8\,\mathrm{m} \times 102.4\,\mathrm{m}$. In contrast, agents equipped with camera sensors have a limited perception range of $[-51.2\,\mathrm{m},\ 51.2\,\mathrm{m}]$ along both axes (i.e., $102.4\,\mathrm{m} \times 102.4\,\mathrm{m}$). During testing and inference, the perception range is unified for all agents to $[-102.4\,\mathrm{m},\ 102.4\,\mathrm{m}]$ along the $x$-axis and $[-51.2\,\mathrm{m},\ 51.2\,\mathrm{m}]$ along the $y$-axis.

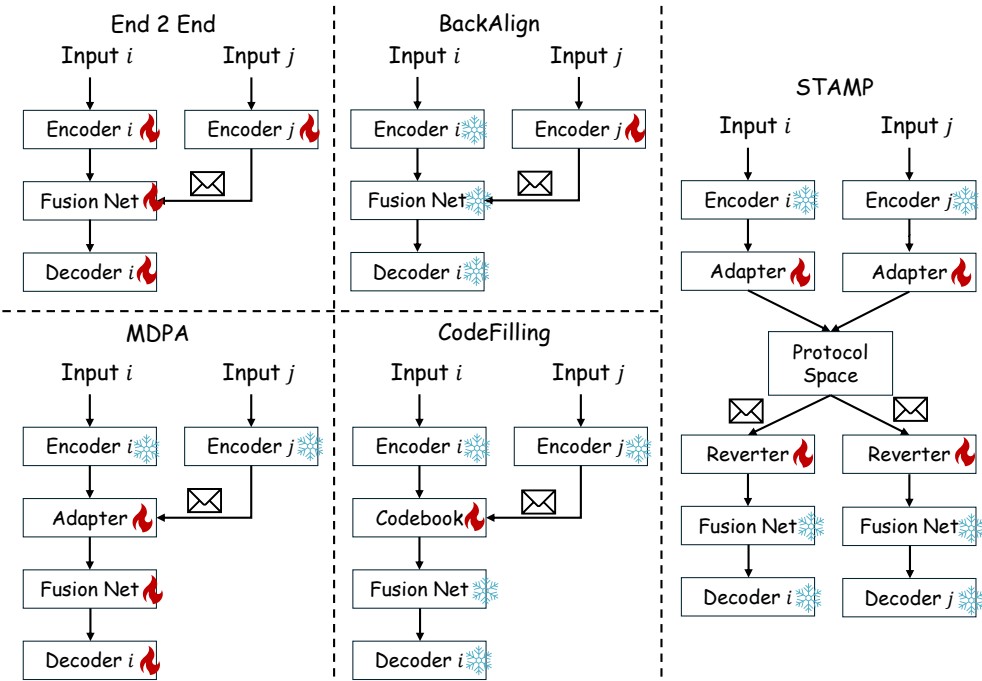

Figure A.2: Illustration of the training strategies used in baseline methods when new heterogeneous agents join the collaboration.

The intermediate feature maps have spatial dimensions of $[C, H, W] = [128, 64, 128]$, corresponding to the number of channels, height, and width, respectively. For camera-based agents, due to the smaller perception area, the feature map size is reduced to $[128, 64, 64]$.

**GenComm settings** We design the transmitted information to have a shape of $[C', H_j, W_j]$, where $C' = 2$ denotes the number of channels. The height $H_j$ and width $W_j$ are determined dynamically based on the receiving agent's spatial configuration. The diffusion model is configured with a total time step $\mathcal{T} = 3$, and the denoising network $\epsilon_\theta$ is implemented as a U-Net with 2 layers. Within the Channel Enhancer module, the channel dimensions of $\mathcal{F}_{\text{res}}$ and $\mathcal{F}_{\text{conv}}$ are both set to 64.

### A.3.4 Training strategies between methods

**Baseline training.** We illustrate the training strategies of the baseline method in detail in Figure A.2. The pipeline starts with training individual agents in a homogeneous setting, which serves as the foundation for subsequent heterogeneous collaboration. The baseline model is then trained on top of these pretrained models. End-to-end training requires retraining all modules of the heterogeneous agents. BackAlign[12] achieves heterogeneous collaboration by aligning the collaborators' semantic space with that of the ego agent, which necessitates retraining the collaborators' encoders. MPDA[9] introduces a dedicated adapter for each collaborator and requires retraining both fusion network and a decoder. CodeFilling[13] updates the codebook whenever a new collaborator joins, ensuring the codebook includes representations for the new agent. STAMP[11] equips each agent with an adapter that maps its own semantic space to a shared protocol space, as well as a converter that maps the protocol space back to its native space. For all pre-trained model using AttFuse[16] as the fusion network, we train for 20 epochs with an initial learning rate of 0.002, using the Adam[42] optimizer. The learning rate is decayed by a factor of 0.1 at the 10th and 15th epochs. For pre-trained model using V2X-ViT[17] as the fusion network, training is conducted for 30 epochs with the same initial learning rate, which is decayed by 0.1 at the 15th and 20th epochs. Based on the pretrained base models, BackAlign, MPDA, and CodeFilling are fine-tuned for 10 additional epochs with an initial learning rate of 0.001, and the learning rate is decayed by a factor of 0.1 at epoch 5. For STAMP, we follow its training schedule: the model is fine-tuned for 5 epochs with an initial learning rate of 0.01, and the learning rate is decayed by 0.1 at epochs 1, 3, and 4.

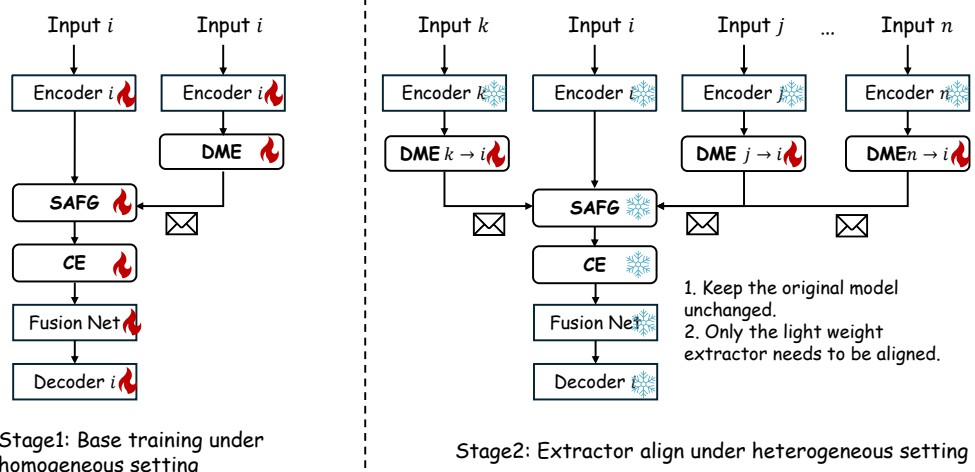

Stage1: Base training under homogeneous setting

Stage2: Extractor align under heterogeneous setting

Figure A.3: Training strategy of GenComm

Table A.3: Robustness analysis on the real-world DAIR-V2X dataset with pose noise. Gaussian noise $\mathcal{N}(0, \sigma_l^2)$ is added to the $x$ and $y$ positions, and $\mathcal{N}(0, \sigma_y^2)$ to the yaw angle.

| Noise level ($\sigma_l^2/\sigma_y^2$) | | 0.0/0.0 | 0.1/0.1 | 0.2/0.2 | 0.3/0.3 | 0.4/0.4 |
|---|---|---|---|---|---|---|
| Methods / Metrics | | | | AP30 ↑ | | |
| AttFuse[16] | MPDA[9] | 0.4296 | 0.4282 | 0.4214 | 0.3964 | 0.3522 |
| | BackAlign[12] | 0.4560 | 0.4528 | 0.4435 | 0.4187 | 0.3688 |
| | CodeFilling[13] | 0.3854 | 0.3840 | 0.3750 | 0.3526 | 0.3189 |
| | STAMP | 0.4468 | 0.4449 | 0.4379 | 0.4162 | **0.377** |
| | GenComm | **0.4608** | **0.4586** | **0.4476** | **0.4198** | 0.3726 |
| V2XViT | MPDA[9] | 0.4717 | 0.4708 | 0.4550 | 0.4257 | 0.3825 |
| | BackAlign[12] | 0.4898 | 0.4874 | 0.4710 | 0.4367 | 0.3783 |
| | CodeFilling[13] | 0.4443 | 0.4390 | 0.4266 | 0.3969 | 0.3521 |
| | STAMP | 0.5421 | 0.5396 | 0.5327 | 0.5051 | **0.4586** |
| | GenComm | **0.5624** | **0.5616** | **0.5438** | **0.5075** | 0.4398 |
| Methods / Metrics | | | | AP50 ↑ | | |
| AttFuse | MPDA[9] | 0.3685 | 0.3591 | 0.3170 | 0.2372 | 0.1624 |
| | BackAlign[12] | 0.3725 | 0.3628 | 0.3181 | 0.2374 | 0.1652 |
| | CodeFilling[13] | 0.3185 | 0.3099 | 0.2697 | 0.2021 | 0.1495 |
| | STAMP | **0.3913** | **0.3817** | **0.3383** | **0.2582** | **0.1899** |
| | GenComm | 0.3801 | 0.3681 | 0.3195 | 0.2415 | 0.1653 |
| V2XViT[17] | MPDA[9] | 0.3779 | 0.3674 | 0.3178 | 0.2377 | 0.1702 |
| | BackAlign[12] | 0.3940 | 0.3819 | 0.3253 | 0.2359 | 0.1591 |
| | CodeFilling[13] | 0.3559 | 0.3483 | 0.2920 | 0.2161 | 0.1504 |
| | STAMP | **0.4935** | **0.4822** | **0.4266** | **0.3275** | **0.2317** |
| | GenComm | 0.4649 | 0.4534 | 0.3875 | 0.2913 | 0.1963 |

**GenComm training.** As illustrated in Figure A.3, we present the training pipeline of GenComm. In the first stage, the three key components of our method, namely the Deformable Message Extractor (DME), the Spatial-Aware Feature Generator (SAFG), and the Channel Enhancer (CE), are trained in a homogeneous setting. After this stage, only the collaborator-specific Extractor is trained to numerically align the transmitted messages with those of the receiver. This lightweight alignment strategy enables efficient adaptation to newly joined agents at minimal cost, thereby supporting *pragmatic heterogeneous collaboration*. The training hyperparameters for both stages are consistent with those used in the baseline setup.

Table A.4: Comparison of introduced latency across different methods.

| | MPDA | BackAlign | CodeFilling | STAMP | **GenComm** |
|---|---|---|---|---|---|
| **Latency introduced (ms)↓** | 69.918 | 0[*] | 13.390 | 17.934 | 20.7 |

[*] Since BackAlign adopts an encoder alignment strategy without introducing extra modules, its introduced latency is 0 ms.

Table A.5: Performance of GenComm on OPV2V-H when dynamically adding agents.

| Method | $\mathbf{L}_P^{64}$ | | $\mathbf{L}_P^{64} + \mathbf{C}_E$ | | $\mathbf{L}_P^{64} + \mathbf{C}_E + \mathbf{L}_S^{32}$ | | $\mathbf{L}_P^{64} + \mathbf{C}_E + \mathbf{L}_S^{32} + \mathbf{C}_R$ | |
|---|---|---|---|---|---|---|---|---|
| | AP50 ↑ | AP70 ↑ | AP50 ↑ | AP70 ↑ | AP50 ↑ | AP70 ↑ | AP50 ↑ | AP70 ↑ |
| **GenComm** | 0.7381 | 0.5848 | 0.7538 | 0.5951 | 0.7873 | 0.6174 | 0.7876 | 0.6184 |

## A.4 Additional experimental results

### A.4.1 Robustness analysis on DAIR-V2X

We further evaluate the robustness of the proposed method on the real-world DAIR-V2X dataset by simulating pose errors through the addition of Gaussian noise to agent poses. As shown in Table A.3, our method maintains strong robustness under noisy conditions, demonstrating good generalization capability to real-world scenarios.

### A.4.2 Comparison of introduced latency among methods

In autonomous driving scenarios with tight real-time requirements, inference latency is a critical factor that directly impacts system safety and responsiveness. We measured the introduced inference time for all baselines, averaging over 100 iterations after a 10-iteration GPU warm-up to ensure reliability. GenComm's latency is compared to CodeFilling and STAMP and is smaller than the sensor data collection interval (100 ms in OPV2V-H). We consider this latency acceptable and unlikely to affect real-time performance.

### A.4.3 Impact of dynamic agent participation

In real-world deployment, collaborative perception systems are often required to function under dynamic collaboration settings, where participating agents may frequently join or leave the system. In Table 2, we report experiments on OPV2V-H and V2X-Real by **dynamically** introducing additional heterogeneous agents into the system, starting from a single agent and scaling up to four collaborating agents. In Table A.5, We take GenComm on OPV2V-H as an example, the AP70 score is 0.585 with only $\mathbf{L}_P^{64}$. When progressively adding $\mathbf{C}_E$, $\mathbf{L}_S^{32}$, and $\mathbf{C}_R$, the performance increases to 0.595 (+0.010), 0.617 (+0.022), and 0.618 (+0.001), respectively. These results indicate that while adding more agents consistently improves performance, the marginal gains diminish as the collaboration scales up.

**Conversely**, if an agent leaves the collaboration, the performance drop is usually minor, especially when multiple agents remain. Additionally, the performance change depends on the entered or left agent's capability. For example, camera-based $\mathbf{C}_E$ and $\mathbf{C}_R$ contribute less than LiDAR-based $\mathbf{L}_S^{32}$.

Table A.6: Performance under different degradation ratios.

| Degradation Ratio | $\mathbf{L}_H^{128} + \mathbf{L}_L^{128}$ | | $\mathbf{L}_H^{128} + \mathbf{L}_L^{128} + \mathbf{L}_M^{128}$ | | $\mathbf{L}_H^{128} + \mathbf{L}_L^{128} + \mathbf{L}_M^{128} + \mathbf{L}_T^{128}$ | |
|---|---|---|---|---|---|---|
| | AP30↑ | AP50↑ | AP30↑ | AP50↑ | AP30↑ | AP50↑ |
| **0** | **0.685** | **0.618** | **0.696** | **0.630** | **0.714** | **0.636** |
| **0.2** | 0.676 | 0.620 | 0.684 | 0.632 | 0.698 | 0.633 |
| **0.4** | 0.664 | 0.618 | 0.668 | 0.624 | 0.668 | 0.618 |
| **0.6** | 0.650 | 0.613 | 0.646 | 0.610 | 0.638 | 0.603 |
| **0.8** | 0.640 | 0.607 | 0.638 | 0.603 | 0.636 | 0.603 |

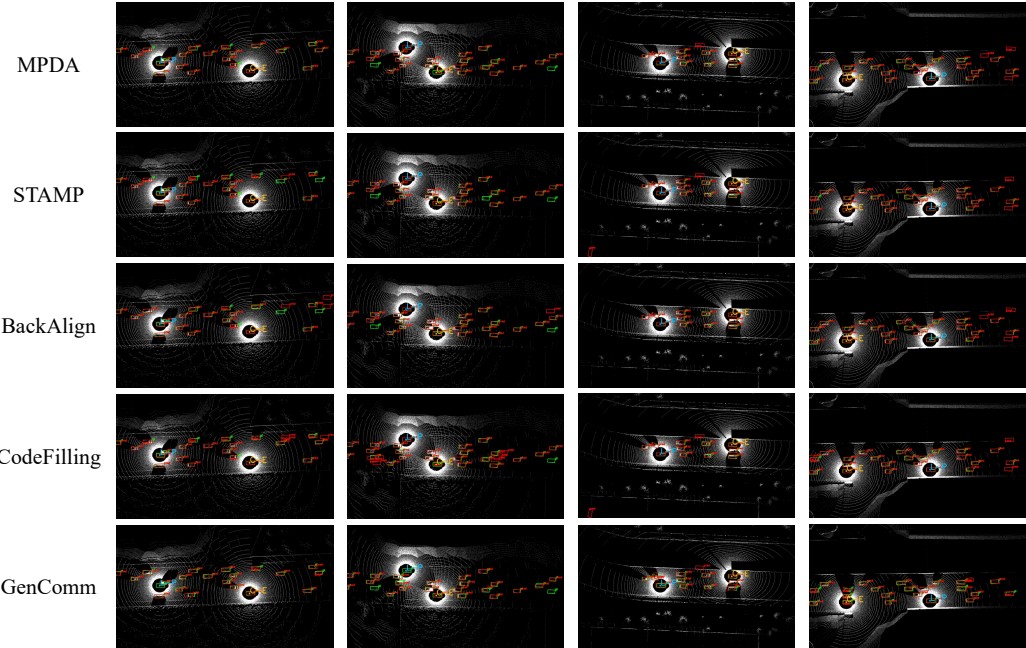

Figure A.4: Visualization of detection results in four different scenarios. The comparisons illustrate that our method delivers more accurate detections with a reduced number of false positives compared to baseline approaches. L-P and C-E, denote the $\mathbf{L}_P^{64}$ and $\mathbf{C}_E$ respectively.

### A.4.4   Impact of degraded information exchange

In practical communication scenarios, factors such as signal interference, bandwidth limitations, and device reliability inevitably lead to partial message degradation during transmission. We further evaluate the robustness of GenComm under message degradation scenarios by randomly applying zero masks with different ratios on the V2X-Real dataset. The corresponding results are presented in Table A.6. However, the degradation keeps in an acceptable range, as the ego agent's perception helps maintain a reasonable lower bound, with AP30 dropping by at most 0.078 and AP50 by 0.033. When the missing ratio is $\leq 0.4$, collaboration still improves performance. When the ratio becomes larger, the messages lose most of their useful information and become closer to noise, potentially harming performance.

### A.5   Visualization

Figure A.4 presents additional visual comparisons of detection results across four different scenarios, demonstrating that our method achieves more precise detections with fewer false positives.

