# OpenReview forum: "Pragmatic Heterogeneous Collaborative Perception via Generative Communication Mechanism"
_NeurIPS.cc/2025/Conference — NeurIPS 2025 poster_

### Official Review · Reviewer_EBDZ · 2025-06-25

**Clarity:** 2
**Significance:** 3
**Originality:** 3
**Rating:** 5
**Confidence:** 5

**Summary:**

This paper proposes GenComm, a generative communication mechanism for heterogeneous collaborative perception. Unlike prior adaptation- or reconstruction-based approaches, GenComm uses a diffusion-based feature generation framework guided by spatial messages extracted via deformable convolutions. It aims to support collaboration across agents with different sensors and models without modifying existing networks. The method introduces lightweight extractor alignment for handling domain discrepancies and demonstrates significant computational and communication cost reductions on OPV2V-H and DAIR-V2X datasets.

**Questions:**

Does the proposed method support, or potentially support, heterogeneous tasks? For example, if one agent is designed for object detection and another for end-to-end driving, would the proposed method support collaboration between these two agents? The authors are not required to experimentally demonstrate this.

**Ethical Concerns:**

["NO or VERY MINOR ethics concerns only"]

**Final Justification:**

The author has addressed most of my questions and concerns, including

W1: Terminology now clearly defined (classification/regression losses)

Q1: Thoughtful theoretical analysis of cross-task support provided

W3: Critical clarification that DMEs are per agent-type, not per-agent (addressing scalability)

The following weakness is partially resolved:

W2: Two-stage deployment strategy and pairwise vs. global consensus distinction are helpful, but fundamental similarity to protocol-based methods remains - both require consensus, just at different granularities

The author believes that this work overall has a solid contribution to collaborative perception. I will suggest to accept the paper.

**Limitations:**

The proposed method assumes a “point-to-point” setting where all agents have a dedicated DME module designed for the ego agent, which may not be ideal in real-world scenarios. The authors need to either address this concern directly or mention it in the limitations section.

**Quality:**

3

**Strengths And Weaknesses:**

Strengths:

	1.	The first generation-based method for heterogeneous collaborative perception.

	2.	The lightweight alignment mechanism enables the integration of new agents with minimal computational overhead (81% reduction).

	3.	GenComm does not require retraining encoders or core modules, preserving each agent’s semantic space integrity.

	4.	Comprehensive experiments on both simulated (OPV2V-H) and real-world (DAIR-V2X) datasets with consistent improvements.

Weaknesses:

	1.	Some important terms are not clearly explained in the main manuscript. For example, the regression loss is neither clearly defined nor easily inferred from the context.

	2.	The advantage of the proposed method over previous protocol-based methods such as PnPDA and STAMP is not clearly demonstrated. While the authors claim that protocol-based methods require consensus on a protocol space, the reviewer believes that the proposed method also requires consensus—though on the feature space of the reference (first) agent. The authors need to clarify this concern.

	3.	The proposed method assumes a “point-to-point” setting where all agents have a dedicated DME module designed for the ego agent, which may not be ideal in real-world scenarios. The authors need to either address this concern directly or mention it in the limitations section.

---

> ### Author Rebuttal · Authors · 2025-07-31
>
> Dear Reviewer,
>
> First and foremost, I would like to extend my sincere gratitude for your time and effort in reviewing the submission and **thank for recognizing our work is efficient, scalable and effective.** Your insightful comments and constructive suggestions are invaluable to refining this work. In the following, we will address your concerns regarding the weaknesses and questions.
>
> > **W1:** Some important terms are not clearly explained in the main manuscript. For example, the regression loss is neither clearly defined nor easily inferred from the context.
> >
>
> Thank you for pointing this, we apologize for the confusion we made when reviewing the paper.
>
> The classification and regression losses in our loss function refer to standard components in 3D object detection. Specifically, the classification loss is used to predict the category of each anchor, and the regression loss predicts the spatial offset from the anchor to the ground truth.
>
>  **We carefully checked any term in paper to eliminate misunderstanding and we will explicitly add the following sentence to the paper and:**
>
> "The classification and regression losses are standard 3D object detection losses, used to predict the category of each anchor and to regress the offset between the anchor and the ground truth object."
>
> > **W2:** The advantage of the proposed method over previous protocol-based methods such as PnPDA and STAMP is not clearly demonstrated. While the authors claim that protocol-based methods require consensus on a protocol space, the reviewer believes that the proposed method also requires consensus—though on the feature space of the reference (first) agent. The authors need to clarify this concern.
> >
>
> Before answering the advantage of GenComm over PnPDA and STAMP, We introduce GenComm’s real-world application rationale in detail:
>
> In real-world deployments, we assume three vendors: $A,B,C$, and five heterogeneous agent types: $A_1,A_2,B_1,B_2,B_3,C$, where each agent refers to a specific combination of sensor and model architecture. Below we describe how GenComm can be practically applied in such scenarios:
>
> **Stage 1: Homogeneous Pre-training.**
>
> Each vendor **independently** trains their agents using GenComm in a homogeneous collaboration setting. For example, vendor $A$ trains agent $A_1$ in collaboration with other instances of $A_1$ using their **private data** belong to vendor $A$ **independently**. In this stage, the fusion network, generation module, and Channel Enhancer are trained effectively under a homogeneous scenarios.
>
> **Stage 2: Heterogeneous Collaboration.**
>
> If two vendors reach a collaboration censensus, they can enable heterogeneous collaboration by training specific Deformable Message Extractors (DMEs) between heterogeneous agents. For instance, if vendors A and B agree to collaborate, they train DMEs such as:$DME_{A_1 \rightarrow B_1},\ DME_{A_1 \rightarrow B_2},\ ..., \ DME_{B_3 \rightarrow A_2}$
>
> These **lightweight** modules are then deployed to the corresponding agents, enabling seamless heterogeneous collaboration without modifying their original backbones. When new agents join the collaboration, vendors simply need to deploy the corresponding new DMEs to the involved agents—**no further modification is required.**
>
> This two-stage process ensures that GenComm is practical, modular, and scalable in real-world multi-agent systems.
>
> Furthermore, the Figure 3 and Figure 4 in supplementary material clearly show the application rational and how the baselines are trained.
>
> Regarding the comparison with protocol-based adaptation methods such as PnPDA and STAMP, we clarify that those approaches require all heterogeneous agents to reach consensus on a shared protocol feature space, which is **challenging in open-world scenarios involving proprietary vendors.** In contrast, GenComm only requires pairwise consensus between vendors to build a specific DMEs for heterogeneous agents, rather than relying on the feature space of the reference (first) agent, as BackAlign does. This significantly relaxes the constraint—reaching consensus between any two vendors is more feasible than achieving global consensus among all vendors, enabling bidirectional communication and better scalability in open environments.
>
>
> > **W3 & Limitations:** The proposed method assumes a “point-to-point” setting where all agents have a dedicated DME module designed for the ego agent, which may not be ideal in real-world scenarios. The authors need to either address this concern directly or mention it in the limitations section.
> >
>
> We have elaborate the details about application rational of GenComm in response to **W2.**
>
> Based on the above, we can infer that this is practical from an implementation perspective, especially since **agents of the same type, meaning those with a specific combination of sensor  and encoder architecture, share same configurations**. It is important to distinguish between **the number of heterogeneous agent types, which is usually limited and tied to vendors**, and **the total number of agents on the road, which can be very large**. The number of DMEs that one agent type needs to maintain depends only on **how many other heterogeneous types it has reached consensus with** and **does not depend on the total number of agents**.
>
> Moreover, each DME is lightweight, containing about **0.078 million parameters**, which does not introduce significant storage or computational overhead.
>
> However, we agree there is a limitation: communication between all heterogeneous types still needs vendors to agree on building DMEs, **which may be affected by competition and intellectual property issues**. We will clearly mention this limitation in the revised paper:
>
> "In real-world applications, although the homogeneous pre-training stage can be independently implemented by each vendor without any privacy leakage concerns, the second stage of building specific DMEs for agent types from other vendors requires pairwise consensus between different vendors."
>
> > **Q1:** Does the proposed method support, or potentially support, heterogeneous tasks? For example, if one agent is designed for object detection and another for end-to-end driving, would the proposed method support collaboration between these two agents? The authors are not required to experimentally demonstrate this.
> >
>
> Thank you for this question, this is an **interesting and thoughtful question that provides valuable insights for both our current work and future research.**
>
> The answer is **Yes**, the proposed method potentially support heterogeneous tasks **theoretically.** The analysis are below:
>
> While we do not directly demonstrate cross-task collaboration in this work, the proposed method is inherently compatible with heterogeneous agents performing different tasks. Since our design focuses on lightweight spatial feature sharing, **each agent can independently decode the received message for its own task-specific purpose** (e.g., object detection, end-to-end planning). This flexibility makes our method suitable for real-world multi-agent systems with specialized roles.
>
> For instance, assume a agent $\mathbf{D}$ is trained by a single task 3d object detection, a agent $\mathbf{M}$ is trained by multi-task 3d object detection, segmentation, occupancy prediction. a agent $\mathbf{U}$ is trained by a unifiled end-to-end paradigm like UniAD[1].
>
> The same point of these three agents is the intermediate feature they used are all bev feature map, but contains difference semantic information due to difference suppervise signal. These agents also can share their spatial information to other agents, **the key point is that how to effectively leaverage the recieved message to fusion with their own feature and** **enhance the feature representation.** One possible solution is, add a specific message decoder on receiver, in the GenComm’s application rational is in satge 2, traininig a specific $DME_{i \rightarrow j}$  on agent  $i$ (as the sender), and **a specific message decoder $MD_{i \rightarrow j}$ on agent $j$ (receiver).** If the agent can use the received information to improve its own features, then the collaboration is effective.
>
> **This interesting topic can be explored in future work to investigate how generative communication mechanisms can enable effective collaboration in task-heterogeneous scenarios.**
>
> ### **To summary:**
>
> (i) we have explain the terms
>
> (ii) we give detaled application rational of GenComm to further demonstrate the advantages of GenComm over PnPDA and STAMP.
>
> (iii) We discuss the collaboration between agents with heterogeneous tasks.
>
> We truly appreciate the reviewer’s insightful feedback, and we are confident that the provided analysis helps clarify the concerns and enhances the overall quality of our work.
>
> References
>
> [1] Hu et al., Planning-oriented Autonomous Driving. CVPR 2023

---

### Official Review · Reviewer_E5U6 · 2025-07-02

**Clarity:** 2
**Significance:** 2
**Originality:** 3
**Rating:** 2
**Confidence:** 3

**Summary:**

This paper proposes GenComm, a generation-based framework for collaborative perception in
multi-agent systems. Unlike adaptation- or reconstruction-based methods, GenComm enables
agents to exchange compact spatial messages via a Deformable Message Extractor, which are
then used by a conditional diffusion model to synthesize aligned features at the ego agent. A
Channel Enhancer further refines these features before fusion. This approach preserves
backbone integrity, maintains semantic consistency, and scales efficiently to new agents.
Experiments on OPV2V-H and DAIR-V2X show that GenComm achieves state-of-the-art
performance with lower communication and computation costs

**Questions:**

1. How robust is the generative feature alignment process to inaccuracies in the spatial
message (e.g., due to sensor noise or misalignment)?
2. Can the proposed framework support dynamic collaboration where agents enter/leave
arbitrarily with partial observability (e.g., missing or stale messages)? How does
performance degrade?
3. Could the authors provide empirical latency comparisons for the diffusion-based generation
vs. adaptation-based baselines?
4. The homogeneous pretraining stage seems to assume shared access to similar agents. How
realistic is this in open-world scenarios with proprietary vendors?

**Ethical Concerns:**

["NO or VERY MINOR ethics concerns only"]

**Limitations:**

Yes.

**Quality:**

2

**Strengths And Weaknesses:**

Strengths
1. The proposed generation-based paradigm offers a novel and effective alternative to
traditional reconstruction approaches, enabling scalable heterogeneous collaboration.
2. Extensive experiments on both synthetic and real-world datasets demonstrate superior
performance and significant cost savings.

Weaknesses
1. Real-world evaluation is limited: DAIR-V2X involves only two agents, which restricts claims
about large-scale applicability.
2. The diffusion component, while effective, may introduce latency concerns in real-time
settings; inference efficiency is not quantified.
3. The framework assumes a homogeneous pretraining stage, which may not be feasible in
decentralized or privacy-restricted environments.

---

> ### Author Rebuttal · Authors · 2025-07-31
>
> Dear Reviewer,
>
> We truly appreciate your thoughtful review and **thank for recognizing our method is scalable, achieves superior performance, and offers significant cost savings.** Your comments and valuable feedback have been instrumental in helping us improve our work. In the following, we respond to your concerns and clarify the relevant points.
>
> > **W1:** Real-world evaluation is limited: DAIR-V2X involves only two agents, which restricts claims about large-scale applicability.
> >
>
> Thank you for pointing this out. We agree that conducting experiments solely on the real-world DAIR-V2X dataset, which contains only two agents, is insufficient to support our claim of large-scale applicability. Therefore, **we extended our experiments to another real-world dataset, V2X-Real [1]**, which includes **four agents.** We evaluated four different agent types: $\mathbf{L}_H^{128},\ \mathbf{L}_L^{64},\ \mathbf{L}_M^{128},\ \mathbf{L}_T^{64}$(Specifically, $\mathbf{L}$ denotes LiDAR modality, 128 indicates the number of LiDAR beams, and the subscripts $H, L, M,$ and $T$ refer to different encoder scales.). The results are below.
>
> |  | $\mathbf{L}_H^{128}+\mathbf{L}_L^{64}$ | $\mathbf{L}_H^{128}+\mathbf{L}_L^{64} + \mathbf{L}_M^{128}$ | $\mathbf{L}_H^{128}+\mathbf{L}_L^{64} + \mathbf{L}_M^{128} + \mathbf{L}_T^{64}$ |
> | --- | --- | --- | --- |
> |  | **AP30/AP50** | **AP30/AP50** | **AP30/AP50** |
> | BackAlign | 0.631/0.582 | 0.638/0.587 | 0.635/0.589 |
> | MPDA | 0.634/0.572 | 0.632/0.575 | 0.621/0.567 |
> | CodeFilling | 0.627/0.582 | 0.628/0.58 | 0.608/0.557 |
> | **GenComm** | **0.685/0.618** | **0.696/0.63** | **0.714/0.636** |
>
> *Note：STAMP needs much longer training time, we will put the complement experiments in the final version if the paper is accepted.*
>
> We see that GenComm outperforms baselines on V2X-Real. **We believe these newly added experiments further demonstrate GenComm’s scalability and potential for real-world applicability.** We will revise the conclusion make it more reasonable:
>
> “These advantages make GenComm a scalable and **potential solution** for real-world multi-agent systems deployment.”
>
> > **W2:** The diffusion component, while effective, may introduce latency concerns in real-time settings; inference efficiency is not quantified.
> >
>
> **We evaluated the inference latency of the GenComm module under different numbers of spatial message channels in ablation study.** From Figure 4 in paper, we see that the inference time of GenComm is 20.7 ms in our experiment setting, which is significantly smaller than the sensor sampling interval of 100 ms. Therefore, **we consider this latency acceptable and unlikely to hinder real-time performance.**
>
> At the same time, we understand the reviewer’s concern that reporting only GenComm’s latency is not convincing. **To address this, we have additionally measured the extra inference time introduced by all baselines.**  Please see details in response to **Q3.**
>
> > **W3:** The framework assumes a homogeneous pre-training stage, which may not be feasible in decentralized or privacy-restricted environments.
> >
>
> To clarify the concern, we need clearly explain the GenComm’s Real-World Application Rational first (Figure 4 in supplementary):
>
> In real-world deployments, we assume three vendors: $A,B,C$, and five heterogeneous agent types: $A_1,A_2,B_1,B_2,B_3,C$, **where each agent type refers to a specific combination of sensor and model architecture.** Below we describe how GenComm can be practically applied in such scenarios:
>
> **Stage 1: Homogeneous Pre-training.**
>
> Each vendor **independently** trains their agents using GenComm in a homogeneous collaboration setting. For example, vendor $A$ trains agent $A_1$ in collaboration with other instances of $A_1$ using their **private data** belong to vendor $A$ **independently**. In this stage, the fusion network, generation module, and Channel Enhancer are trained effectively under a homogeneous scenarios.
>
> **Stage 2: Heterogeneous Collaboration.**
>
> If two vendors reach a collaboration censensus, they can enable heterogeneous collaboration by training specific Deformable Message Extractors (DMEs) between heterogeneous agents. For instance, if vendors A and B agree to collaborate, they train DMEs such as:$DME_{A_1 \rightarrow B_1},\ DME_{A_1 \rightarrow B_2},\ ..., \ DME_{B_3 \rightarrow A_2}$
>
> These lightweight modules enable heterogeneous collaboration by deploying correspond DMEs to each agent. When new agents join, vendors only need to deploy the involved DMEs **without further changes.**
>
> Now we clarify that each vendor can perform stage 1 **independently with their own agent types and private data. Therefore, this stage involves no cross-vendor data or model sharing and raises no privacy concerns.**
>
> In response to **Q4,** we give more clarification about privacy concern about real-world application.
>
> > **Q1:** How robust is the generative feature alignment process to inaccuracies in the spatial message (e.g., due to sensor noise or misalignment)?
> >
>
> We evaluated **robustness** on OPV2V-H and DAIR-V2X (Figure 3 in paper, Table 3 in supplementary material), testing **pose noise (0–0.8 std)** and **latency (0–500 ms)** that cause **spatial misalignment**. **GenComm consistently outperforms baselines**, showing robustness to imperfect, noisy inputs in realistic settings.
>
> > **Q2:** Can the proposed framework support dynamic collaboration where agents enter/leave arbitrarily with partial observability (e.g., missing or stale messages)? How does performance degrade?
> >
>
> In Table 2 of the main paper and response to **W1**, we present experiments on OPV2V-H and V2X-Real by **dynamically** introducing additional heterogeneous agents into the system one by one, starting from a single agent and scaling up to four collaborating agents. Taking GenComm on OPV2V-H as an example, the AP70 is 0.585 with only $\mathbf{L}_P^{64}$. Adding $\mathbf{C}_E, \mathbf{L}_S^{32}$, and $\mathbf{C}_R$ increases it to 0.595 (**+0.010**), 0.617 (**+0.022**), and 0.618 (**+0.001**), showing improvements but only marginal gains as more agents join.
>
> | **Method** | $\mathbf{L}_P^{64}$ | $\mathbf{L}_P^{64}+\mathbf{C}_E$ | $\mathbf{L}_P^{64}+\mathbf{C}_E+\mathbf{L}_S^{32}$ | $\mathbf{L}_P^{64}+\mathbf{C}_E+\mathbf{L}_S^{32}+\mathbf{C}_R$ |
> | --- | --- | --- | --- | --- |
> |  | **AP50/AP70** | **AP50/AP70** | **AP50/AP70** | **AP50/AP70** |
> | **GenComm** | 0.738/0.585 | 0.754/0.595 | 0.787/0.617 | 0.787/0.618 |
>
> **Conversely**, If an agent leaves the collaboration, the performance drop is usually minor, especially when multiple agents remain. Additionally, the performance change depends on the entered or left agent’s capability. For example, camera-based $\mathbf{C}_E$ and $\mathbf{C}_R$ contribute less than LiDAR-based $\mathbf{L}_S^{32}$.
>
> We also conducted experiments simulating missing message scenarios by applying random zero mask of different ratios. The results are shown below:
>
> | **Missing_ratio** | $\mathbf{L}_H^{128}+\mathbf{L}_L^{64}$ | $\mathbf{L}_H^{128}+\mathbf{L}_L^{64} + \mathbf{L}_M^{128}$ | $\mathbf{L}_H^{128}+\mathbf{L}_L^{64} + \mathbf{L}_M^{128} + \mathbf{L}_T^{64}$ |
> | --- | --- | --- | --- |
> |  | AP30/AP50$\uparrow$ | AP30/AP50$\uparrow$ | AP30/AP50$\uparrow$ |
> | **0** | **0.685/0.618** | **0.696/0.63** | **0.714/0.636** |
> | **0.2** | 0.676/0.62 | 0.684/0.632 | 0.698/0.633 |
> | **0.4** | 0.664/0.618 | 0.668/0.624 | 0.668/0.618 |
> | **0.6** | 0.650/0.613 | 0.646/0.610 | 0.638/0.603 |
> | **0.8** | 0.640/0.607 | 0.638/0.603 | 0.636/0.603 |
>
> Overall, missing information causes a performance drop. **However, the degradation keep in an acceptable range, as the ego agent’s perception helps maintain a reasonable lower bound**, with AP30 dropping by at most 0.078 and AP50 by 0.033.
>
> When the missing ratio is ≤ 0.4, collaboration still improves performance. When the ratio becomes larger, the messages lose most of their useful information and closer to noise, potentially harming performance.
>
> > **Q3:** Could the authors provide empirical latency comparisons for the diffusion-based generation vs. adaptation-based baselines?
> >
>
> We measured the additional inference time for all baselines, averaging over 100 iterations after a 10-iteration GPU warm-up to ensure reliability.
>
> |  | MPDA | BackAlign | CodeFilling | STAMP | **GenComm** |
> | --- | --- | --- | --- | --- | --- |
> | **Latency introduced(ms)$\downarrow$** | 69.918 | 0 | 13.390 | 17.934 | 20.7 |
>
> GenComm’s latency is compared to CodeFiling and STAMP and is smaller than the sensor collection interval (100ms). **We consider this latency acceptable and unlikely to affect real-time performance.**
>
> > **Q4:** The homogeneous pretraining stage seems to assume shared access to similar agents. How realistic is this in open-world scenarios with proprietary vendors?
> >
>
> In response to **W3**, we have provided detailes on GenComm’s real-world applicability and clarified that the homogeneous pre-training stage can be independently conducted by each vendor without raising privacy concerns.
>
> **We further discuss Stage 2 in open-world scenarios with proprietary vendors.**  Collaboration occurs only when both vendors reach on a consensus in **privacy policies.** In contrast to protocol-based adaptation approaches such as PnPDA and STAMP, which require all heterogeneous agents to adopt a unified protocol feature space, or methods like Codebook and MPDA that assume joint training across all heterogeneous agents, or BackAlign which forces other vendors to align with the ego agent’s feature space. **GenComm offers a more realistic and scalable alternative that is both practical and privacy-preserving.**
>
> At the end, we are grateful for the reviewer’s valuable suggestions and believe that our additional experiments and analysis has effectively resolved the raised concerns and improved the submission.
>
> Reference
>
> [1] Xiang et al., V2X-REAL: A Large-Scale Dataset for Vehicle-to-Everything Cooperative Perception. ECCV 2024

---

### Official Review · Reviewer_Ltr8 · 2025-07-02

**Clarity:** 3
**Significance:** 2
**Originality:** 4
**Rating:** 5
**Confidence:** 3

**Summary:**

This paper presents a novel approach to facilitate multi-agent communication that optimizes over communication capacity and computing power needed to adapt to new agents. It has clear real world applications in scenarios like multi-vehicle autonomous driving. Experiments show the method's effectiveness in reducing the communication capacity requirement.

**Questions:**

1. How is the multi-agent performance in terms of pairwise communication volume, etc.? Will it work among multiple settings on more than 2 agents, e.g., a clique-like communication graph or a centralized dispatcher or anything else that is meaningful in real world communication?
2. The problem itself appears to be decentralized. Are there ways / metrics to evaluate the consistency among different agent's decoding results?

I will consider increasing my score if the above concerns are addressed properly.

**Ethical Concerns:**

["NO or VERY MINOR ethics concerns only"]

**Final Justification:**

The authors have addressed most of my concerns except when I was asking about evaluating consistency I was actually expecting some results of those metrics. That is my problem of course for not stating it clearly in the review so I won't take that into final reviews of the paper. I decide to increase my score to 5 in answer to the authors' response of new experiments in multi-agent and real world settings.

**Limitations:**

yes

**Quality:**

3

**Strengths And Weaknesses:**

Strengths:
The paper's main result is comprehensively evaluated and seems effective. The diffusion based multi agent perception has been proven useful in other settings, so it's not surprising to see it working in the autonomous driving environment, but it's interesting to see it actually also reduces communication volume.

Weaknesses:
There's not much evidence in the experiments part that the multi-agent setting is working well with more than 2 agents.

---

> ### Author Rebuttal · Authors · 2025-07-31
>
> Dear Reviewer,
>
> Thank you sincerely for taking the time to review our work and **for recognizing the effectiveness and efficiency of GenComm.** We greatly value your constructive feedback and insightful comments, which are essential to enhancing the quality of our paper. Below, we carefully address the concerns and questions you have raised.
>
> > **W1:** There's not much evidence in the experiments part that the multi-agent setting is working well with more than 2 agents.
> >
>
> In response to the weakness, we provide further evidence showing that GenComm works effectively with more than two agents.
>
> In Table 2 of the paper, we evaluate our method on the simulation dataset OPV2V-H using **four heterogeneous agents in collaboration** :
>
> 1.Agent $\mathbf{L}_P^{64}$ using LiDAR modality with 64 beams sensor and **Point Pillar** encoder；
>
> 2.Agent $\mathbf{C}_E$ using camera modality with **EfficientNet** encoder;
>
> 3.Agent $\mathbf{L}_S^{32}$ using LiDAR modality with 32 beams sensor and **SECOND** encoder;
>
> 4.Agent $\mathbf{C}_R$ using camera modality with **ResNet** encoder.
>
> As shown in the results, the overall performance consistently improves with the addition of each new agent. This demonstrates that our method **scales well and works effectively** in scenarios involving more than two agents.
>
> We will change the table header of Table 2 like below to clearly show that the total agent number is increasing:
>
> | Method/Metric | $\mathbf{L}_P^{64}+\mathbf{C}_E$ | $\mathbf{L}_P^{64}+\mathbf{C}_E+\mathbf{L}_S^{32}$ | $\mathbf{L}_P^{64}+\mathbf{C}_E+\mathbf{L}_S^{32}+\mathbf{C}_R$ |
> | --- | --- | --- | --- |
> |  | AP50/AP70 | AP50/AP70 | AP50/AP70 |
> | **GenComm** | 0.754/0.595 | 0.787/0.617 | 0.787/0.618 |
>
> While DAIR-V2X is a real-world dataset, it does not fully demonstrate GenComm's effectiveness with more than two agents in real-world settings. To address this, we **additionally introduce another real-world dataset V2X-REAL[1],** which contains **four agents**. We conduct experiments on V2X-REAL to further demonstrate that our model outperforms existing methods in real-world settings with more than two agents. We evaluated four different agent types: $\mathbf{L}_H^{128},\ \mathbf{L}_L^{64},\ \mathbf{L}_M^{128},\ \mathbf{L}_T^{64}$(Specifically, $\mathbf{L}$ denotes LiDAR modality, 128 indicates the number of LiDAR beams, and the subscripts $H, L, M,$ and $T$ refer to different encoder scales.). The results are below.
>
> |  | $\mathbf{L}_H^{128}+\mathbf{L}_L^{64}$ | $\mathbf{L}_H^{128}+\mathbf{L}_L^{64} + \mathbf{L}_M^{128}$ | $\mathbf{L}_H^{128}+\mathbf{L}_L^{64} + \mathbf{L}_M^{128} + \mathbf{L}_T^{64}$ |
> | --- | --- | --- | --- |
> |  | **AP30/AP50$\uparrow$** | **AP30/AP50$\uparrow$** | **AP30/AP50$\uparrow$** |
> | **BackAlign** | 0.631/0.582 | 0.638/0.587 | 0.635/0.589 |
> | **MPDA** | 0.634/0.572 | 0.632/0.575 | 0.621/0.567 |
> | **CodeFilling** | 0.627/0.582 | 0.628/0.58 | 0.608/0.557 |
> | **GenComm** | **0.685/0.618** | **0.696/0.63** | **0.714/0.636** |
>
> *Note：STAMP needs much longer training time, we will put the complement experiments in the final version if the paper is accepted.*
>
> We see that GenComm outperforms existing methods on V2X-Real. **We believe these newly added experiments further demonstrate GenComm is working well with more than 2 agents on real-world dataset.**
>
> > **Q1:** How is the multi-agent performance in terms of pairwise communication volume, etc.? Will it work among multiple settings on more than 2 agents, e.g., a clique-like communication graph or a centralized dispatcher or anything else that is meaningful in real world communication?
> >
>
> GenComm still maintains low communication volume even in settings with more than two agents. We provide a detailed analysis below:
>
> Firstly, we report the **communication volume per single transmission** (indicates that the collaborator sends one message to the receiver during a single communication round.) for GenComm and baseline methods in Table 1 of the paper, the result as shown below:
>
> |  | MPDA | BackAlign | CodeFilling | STAMP | **GenComm** |
> | --- | --- | --- | --- | --- | --- |
> | Comm. Volume($log_2$ scale) $\downarrow$ | 22.0 | 22.0 | 15.0 | 22.0 | **16.0** |
>
> In our experimental setting, GenComm achieves a **64× reduction in Communication volume** per transmission compared to MPDA, BackAlign, and STAMP. As the number of agents increases, the number of communication times increases accordingly. Given a fixed per transmission volume, the total communication cost grows linearly with the number of transmission times. Therefore, the bandwidth-saving advantage of GenComm **naturally extends to multi-agent scenarios beyond two agents.**
>
> Next, we consider several practical communication structures to further analyze the communication volume under different system:
>
> - Clique-like communication graph: Consider a clique-like communication graph consisting of $N$ agents. In this setting, each agent simultaneously acts as both a receiver (receiving messages) and a collaborator (sending messages). Consequently, the total number of communications amounts to  $N×(N+1)$.
> - Centralized dispatcher: Assume a cloud server or a designated agent acts as a centralized dispatcher that receives, processes, and forwards messages to the other $N$ agents. In this scenario, the total number of communications is  $2\times N$.
>
> Before introducing GenComm’s communication topology, we first elaborate on its real-world application rationale (Presented as an image in Figure 4 of the supplementary material.):
>
> In real-world deployments, we assume three vendors: $A,B,C$, and five heterogeneous agent types: $A_1,A_2,B_1,B_2,B_3,C$, **where each agent type refers to a specific combination of sensor and model architecture.** Below we describe how GenComm can be practically applied in such scenarios:
>
> **Stage 1: Homogeneous Pre-training.**
>
> Each vendor **independently** trains their agents using GenComm in a homogeneous collaboration setting. For example, vendor $A$ trains agent $A_1$ in collaboration with other instances of $A_1$ using their **private data** belong to vendor $A$ **independently**. In this stage, the fusion network, generation module, and Channel Enhancer are trained effectively under a homogeneous scenarios.
>
> **Stage 2: Heterogeneous Collaboration.**
>
> If two vendors reach a collaboration consensus, they can enable heterogeneous collaboration by training specific Deformable Message Extractors (DMEs) between heterogeneous agents. For instance, if vendors A and B agree to collaborate, they train DMEs such as:$DME_{A_1 \rightarrow B_1},\ DME_{A_1 \rightarrow B_2},\ ..., \ DME_{B_3 \rightarrow A_2}$
>
> These lightweight modules enable heterogeneous collaboration by deploying correspond DMEs to each agent. When new agents join, vendors only need to deploy the involved DMEs **without further changes.**
>
> From the above, GenComm’s real-world communication structure is as follows:
>
> - GenComm Real-world Application Rational: Agents of different types from various vendors can participate in collaboration once **pairwise consensus** is established between vendors. Instead of assuming a fully connected (clique-like) communication topology, GenComm adopts a **non-fully connected communication graph**, which is more realistic for real-world deployment. In this setting, the total communication volume depends on how many collaborators each agent connects with. Suppose there are $N$ agents $V_1, V_2, ..., V_N$, and agent $V_i$ communicates with $W_i$collaborators (where $W_i \leq N - 1$). The total number of communication times is $\sum_{i=1}^{N} W_i$. This design enables scalable and efficient collaboration with limited bandwidth requirements in real-world application.
>
> > **Q2:** The problem itself appears to be decentralized. Are there ways / metrics to evaluate the consistency among different agent's decoding results?
> >
>
> Yes, we provide our insights about the metrics to evaluate the consistency among different agent's decoding results.
>
> For a single agent, decoding results depend on the specific downstream task. For example, 3D object detection produces bounding boxes, semantic segmentation yields pixel-level labels, and occupancy prediction outputs binary occupancy values. Each agent can independently decode and evaluate results based on its own task and local ground truth.
>
> In perception stage (what we focus in this paper),  for the same task, **using the same evaluation metrics(mAP for 3d object detection, mIoU for segmentation) can provide a reasonable indication of consistency among agents’ decoding results.** This is because, when evaluating within the same collaboration scene, the ground truth used by all agents is identical. For instance, in a 3D object detection task, all agents use the same ground-truth bounding boxes. Similarly, in an instance segmentation task, the segmentation maps are the same. Further, **the Intersection-over-Union (IoU) of bounding boxes or segmentation maps from different agents can also be used to directly evaluate consistency.**
>
> Additionally, consistency among agents can also be evaluated at the prediction and decision-making stages. Specifically, we can assess **whether the collaborative decisions made after multi-agent fusion converge to the same or similar conclusions.**
>
> ### **To summary:**
>
> **(i)** We provide evidence and add experiments to show that GenComm works well with more than two agents.
>
> **(ii)** We provide a detailed application rationale for GenComm and analyze the communication volume in several cases.
>
> **(iii)** We discuss how to evaluate result consistency between heterogeneous agents.
>
> Thank you for the constructive feedback. We believe our newly added experiments and analysis responds well to the concerns and contributes to a better version of our paper.
>
> References
>
> [1] Xiang et al., V2X-REAL: A Large-Scale Dataset for Vehicle-to-Everything Cooperative Perception. ECCV 2024

---

> > ### Comment · Reviewer_Ltr8 · 2025-08-05
> >
> > Thanks to the authors for the prompt response! The authors have addressed most of my concerns except when I was asking about evaluating consistency I was actually expecting some results of those metrics. That is my problem of course for not stating it clearly in the review so I won't take that into final reviews of the paper. I decide to increase my score to 5 in answer to the authors' response of new experiments in multi-agent and real world settings.

---

> > > ### Author Response · Authors · 2025-08-05
> > > **Thanks for your replay!**
> > >
> > > We sincerely thank the reviewer for carefully reading our response and providing thoughtful comments. We truly appreciate your consideration of raising the score to 5. At the same time, we also apologize for the misunderstanding in our response to Q2. In the revised version of our paper, we will include specific metric results to better evaluate the consistency between multiple agents.
> > >
> > > Once again, thank you for your time, effort, and the valuable feedback, which are crucial to improving the quality of our paper.

---

### Official Review · Reviewer_bQk5 · 2025-07-03

**Clarity:** 3
**Significance:** 3
**Originality:** 3
**Rating:** 4
**Confidence:** 3

**Summary:**

Summary：

The paper presents a novel approach for addressing the challenges in heterogeneous multi-agent collaborative perception, particularly the problem of effectively integrating new agents with minimal computational cost and without modifying existing models. The proposed method, called GenComm, utilizes a generative communication mechanism that allows for seamless collaboration among heterogeneous agents with different sensor modalities and model architectures.

**Questions:**

see the weakness

**Ethical Concerns:**

["NO or VERY MINOR ethics concerns only"]

**Limitations:**

yes

**Quality:**

3

**Strengths And Weaknesses:**

Strength：

The introduction of the Generative Communication mechanism (GenComm) is a novel approach to solving the problem of pragmatic heterogeneous collaboration in multi-agent systems. This method avoids retraining the core components of agents and allows for scalable integration of new agents with minimal computational overhead. One of the key strengths of the paper is its focus on scalability. The paper clearly shows that GenComm allows for the addition of new agents with minimal impact on system performance and computational resources, which is crucial for large-scale, real-world applications.

Weakness：

The symbols \mathcal{M}_{j\rightarrow i} and \mathcal{X}_{i} first appear in equation (1) but are not explained; additionally, there appears to be a typographical error in equation (2e).

Furthermore, the authors claim that GenComm's breakthrough lies in the fact that existing methods (such as feature alignment in BackAlign) rely on architecture adaptation, whereas GenComm avoids model modification through generative feature reconstruction.


Furthermore, while the comparison results showcase performance metrics such as AP50 and AP70, there is no in-depth analysis of why GenComm outperforms these methods. The authors should explore in which specific areas their method demonstrates advantages (e.g., reducing communication costs, decreasing parameter counts, etc.) and discuss how these improvements relate to the limitations of existing work.

---

> ### Author Rebuttal · Authors · 2025-07-31
>
> Dear Reviewer,
>
> We truly appreciate your time and effort in reviewing our submission and sincerely thank **for recognizing our proposed method and its significance for real-world deployment and applications**. Your thoughtful feedback and valuable suggestions play an important role in helping us improve this work. Below, we provide detailed responses to the concerns and questions you raised.
>
> > **W1:** The symbols $\mathcal{M}_{j \rightarrow i}$ and $\mathcal{X}_{i}$ first appear in equation (1) but are not explained; additionally, there appears to be a typographical error in equation (2e).
> >
>
> Thank you for pointing out this issue, and we apologize for the confusion caused during reviewing the paper. **We will add explanations for all symbols before their first appearance, correct all typos, and carefully check all equations and notations to ensure clarity and accuracy.**
>
> "The message of agent $i$  transmitted to agent $j$ is denote as $\mathcal{M}_{i \rightarrow j}$, the observation input of agent $i$ is denote as $ \mathcal{X}_i $"
>
> This sentence will be added to Section 3 and we will revise the formulation (2e) in the revised version as: $\hat{O_i}=f_{dec}^i(\mathcal{Z_i})$
>
> > **W2:** Furthermore, the authors claim that GenComm's breakthrough lies in the fact that existing methods (such as feature alignment in BackAlign) rely on architecture adaptation, whereas GenComm avoids model modification through generative feature reconstruction.
> >
>
> **BackAlign is an intrusive method** that forces collaborators to align their feature space with that of the ego agent, which leads to modifications of the collaborators’ original model parameters. This is unrealistic in real-world applications. In contrast, **GenComm avoids** model modification by employing a generative communication mechanism to bridge the semantic gap between heterogeneous agents. This presents a key advantage of GenComm over BackAlign, especially when incorporating new agents into the system.
>
> > **W3:** Furthermore, while the comparison results showcase performance metrics such as AP50 and AP70, there is no in-depth analysis of why GenComm outperforms these methods. The authors should explore in which specific areas their method demonstrates advantages (e.g., reducing communication costs, decreasing parameter counts, etc.) and discuss how these improvements relate to the limitations of existing work.
> >
>
> Thank you for your comment. We will provide a detailed analysis from both performance and advantage perspectives.
>
> Firstly, the improved performance over the baselines can be mainly attributed to the following reasons:
>
> **(i)** accurate spatial information extraction by the DME (Deformable Message Extractor).
>
> **(ii)** the high-quality generation process.
>
> **(iii)** the channel enhancement introduced by the CE(Channel Enhancer) module.
>
> Together, these components we proposed help **preserve rich semantic information** and **effectively eliminate the semantic gap** introduced by heterogeneous agents better than existing methods, result in improving overall system performance.
>
> We **will add this analysis into Section 5.2** of the Experiments section in the revised paper:
>
> “The performance gain mainly comes from accurate spatial information extraction (DME), high-quality feature generation, and channel enhancement (CE), which together preserve rich semantics and effectively reduce the semantic gap across heterogeneous agents.”
>
> Next, we provide a detailed discussion of the advantages of GenComm compared to existing methods.
>
> GenComm is a generative communication mechanism that generates high-quality feature maps by leveraging lightweight spatial information. **The benefits of this mechanism are three-fold:**
>
> **(i)** It avoids destructing the original network by using a generative approach;
>
> **(ii)** It reduces communication bandwidth by transmitting only compact spatial cues;
>
> **(iii)** It is scalable due to minimal cost to incorporating new agents.
>
> **We can see that GenComm achieves the best performance in terms of communication and computational efficiency,** as shown in Table 2 in our paper:
>
> |  | MPDA | BackAlign | CodeFilling | STAMP | **GenComm** |
> | --- | --- | --- | --- | --- | --- |
> | **#Params(M)$\downarrow$** | 5.75 | 31.18 | 0.81 | 1.64 | **0.31** |
> | **#GFlops(G)$\downarrow$** | 51.93 | 211.38 | 12.91 | 3.084 | **0.615** |
>
> Furthermore, we provide a detailed comparison of our method against existing approaches in terms of key features and advantages, as summarized in the table. The comparison attributes include:
>
> **“Multi-Mod.”**: whether the method has been evaluated with **multi** heterogeneous **mod**alities;
>
> **“Multi-Enc.”**: whether the method has been evaluated with **multi** heterogeneous **enc**oders;
>
> **“Non-Intrus.”**: whether it is **Non**-**intrus**ive, i.e.,  preserves the original model parameters without modifications;
>
> **“Scal.”**: whether the method is **scal**able;
>
> **“Comm. Eff.”**: whether it is **comm**unication-**eff**icient to save bandwidth usage.
>
> | Method | Publication | Multi-Mod. | Multi-Enc. | **Non-Intrus.** | Scal. | Comm. Eff. |
> | --- | --- | --- | --- | --- | --- | --- |
> | MPDA | ICRA 2023 |  | ✓ |  |  |  |
> | HM-ViT | ICCV 2023 | ✓ |  |  |  |  |
> | HEAL | ICLR 2024 | ✓ | ✓ |  |  |  |
> | CodeFilling | CVPR 2024 | ✓ |  |  |  | ✓ |
> | PnPDA | ECCV 2024 |  | ✓ | ✓ |  |  |
> | STAMP | ICLR 2025 | ✓ | ✓ | ✓ | ✓ |  |
> | **GenComm** | - | ✓ | ✓ | ✓ | ✓ | ✓ |
>
> **GenComm demonstrates clear advantages over existing methods in terms of both practicality and scalability, making it a potential solution for real-world multi-agent systems deployment.** These advantages are difficult to achieve simultaneously with existing methods, and we believe they are crucial for advancing the deployment of multi-agent systems in real-world applications.
>
> ### **To summary:**
>
> **(i)** We will carefully check for typos and notation throughout the paper.
>
> **(ii)** We have added an analysis of the performance gain in GenComm.
>
> **(iii)** We provide a comprehensive comparison of GenComm with existing methods.
>
> We sincerely appreciate the reviewer's constructive suggestions and believe that the additional analysis addresses the concerns and significantly improves the quality of our submission.

---

> ### Author Response · Authors · 2025-08-05
>
> We sincerely thank the reviewer for the constructive and thoughtful feedback. In our response, we have made every effort to address your comments and suggestions:
>
> - We will thoroughly check the paper for any typos and inconsistencies in notation.
> - We have added a detailed analysis of the performance gains achieved by GenComm.
> - We have included a comprehensive comparison between GenComm and existing approaches to better highlight its advantages.
>
> We hope these clarifications address your concerns, and we would greatly appreciate any further comments or suggestions you may have. We are happy to continue the discussion and provide additional information if needed.

---

### Author Response · Authors · 2025-08-09
**Overall Comment & GenComm’s real-world significance**

We sincerely appreciate all the reviewers’ time and insightful comments. We have carefully addressed all concerns in our response through additional experiments and detailed clarifications. Furthermore, we hope the reviewers are able to recognize **the significance of our approach and its potential impact on the deployment of heterogeneous collaborative perception systems in real-world scenarios.**

**GenComm** is a generative communication mechanism that generates high-quality feature maps by leveraging lightweight spatial information. Compared to previous methods, this mechanism **simultaneously** meets three-fold benefits:

- **Non-intrusive to existing models:** GenComm adopts a generative approach, thereby avoiding any structural modifications to the original network.
- **Reduced communication bandwidth:** By transmitting only compact spatial cues, GenComm significantly lowers the communication overhead compared to feature or raw-data transmission.
- **Scalable:** The minimal cost of incorporating new agents makes GenComm readily adaptable to emerging heterogeneous agents.

Furthermore, the two-stage application rationale ensures privacy between vendors while providing a deployment setting that is **more realistic** than existing methods.

Altogether, these advantages make GenComm a potential and practical solution for real-world V2X applications.

---

### Note · Authors · 2025-08-12

We sincerely thank all reviewers and the AC for your time and effort. We truly appreciate the reviewers’ recognition of **GenComm** as a novel, effective, bandwidth-efficient, and scalable method for real-world applications. We firmly believe that our approach has much greater potential as a solution for heterogeneous collaborative perception in real-world scenarios than existing methods. In our rebuttal and discussions, we have addressed the main concerns as follows:

- Added new experiments on **V2X-Real** (4 agents) to address *W1* from Reviewer Ltr8 and *W1* from Reviewer E5U6.
- Provided **latency comparisons** between our method and baselines, as well as the results of **missing-message** setting, as requested in *Q2* and *Q3* by Reviewer E5U6.
- Clarified that robustness experiments and latency results of our method are already included in the main paper and supplementary material, as noted by Reviewer E5U6 in *W2* and *Q1*.
- Explained the **application rationale** in detail to address the privacy concerns raised by Reviewer E5U6, showing that our approach is more realistic than existing methods.
- Addressed the **communication paradigm in real-world** scenarios with Reviewer Ltr8 in our response, and engaged in further discussions on **future work** with Reviewer EBDZ.

We will incorporate these valuable and thoughtful comments to further improve our paper:

- Carefully check all notations, terms, table headers and statement to avoid misunderstanding.
- Include a more comprehensive performance analysis of GenComm.
- Clearly describe the application rationale with an illustrative figure for better understanding.
- Integrate all newly added experiments into the final version.

Furthermore, we have provided a detailed description of the real-world significance of **GenComm** in the comment *“Overall Comment & GenComm‘s real-world significance”*. We would be grateful if all reviewers and the AC could take this final remark into consideration.

---

### Decision · Program_Chairs · 2025-09-17

**Decision:**

Accept (poster)

**Comment:**

This paper proposes a new multi-agent communication method, GenComm, a generative communication mechanism for heterogeneous collaborative perception. In particular, GenComm uses a diffusion-based feature generation framework guided by spatial messages extracted via deformable convolutions. It does not need to alter the original network and employs lightweight numerical alignment of spatial information to efficiently integrate new agents at minimal cost. Evaluations on simulated and real-world driving datasets validate the performance of the method.

The rebuttal provided new experiments on V2X-Real with 4 agents, a latency comparison, and provided additional clarification/discussions on results, application rationale, and the communication paradigm in real-world scenarios. All but one reviewers recommended accept or borderline accept. The only reviewer who has a negative rating did not engage in discussion. Considering the rebuttal and the discussions, the AC agrees with the majority of the reviewers and would recommend accept.